# Monoallelically expressed noncoding RNAs form nucleolar territories on NOR-containing chromosomes and regulate rRNA expression

Qinyu Hao[1†], Minxue Liu[1†], Swapna Vidhur Daulatabad[2], Saba Gaffari[3], You Jin Song[1], Rajneesh Srivastava[2], Shivang Bhaskar[1], Anurupa Moitra[1], Hazel Mangan[4], Elizabeth Tseng[5], Rachel B Gilmore[6], Susan M Frier[7], Xin Chen[8], Chengliang Wang[9], Sui Huang[10], Stormy Chamberlain[6], Hong Jin[8,9,11], Jonas Korlach[5], Brian McStay[4], Saurabh Sinha[3,12], Sarath Chandra Janga[2], Supriya G Prasanth[1,13], Kannanganattu V Prasanth[1,13]*

[1]Department of Cell and Developmental Biology, University of Illinois at Urbana-Champaign, Urbana, United States; [2]Department of BioHealth Informatics, School of Informatics and Computing, IUPUI, Indianapolis, United States; [3]Department of Computer Science, University of Illinois at Urbana-Champaign, Urbana, United States; [4]Centre for Chromosome Biology, School of Natural Sciences, National University of Ireland Galway, Galway, Ireland; [5]Pacific Biosciences, Menlo Park, United States; [6]Department of Genetics and Genome Sciences, University of Connecticut School of Medicine, Farmington, United States; [7]Ionis Pharmaceuticals Inc, Carlsbad, United States; [8]Department of Biophysics and Quantitative Biology, University of Illinois at Urbana-Champaign, Urbana, United States; [9]Department of Biochemistry, University of Illinois at Urbana-Champaign, Urbana, United States; [10]Department of Cell and Molecular Biology, Northwestern University, Chicago, United States; [11]Carl R. Woese Institute for Genomic Biology, University of Illinois at Urbana-Champaign, Urbana, United States; [12]Department of Biomedical Engineering, Georgia Tech, Atlanta, United States; [13]Cancer Center at Illinois, University of Illinois at Urbana-Champaign, Urbana, United States

*For correspondence:
kumarp@illinois.edu

†These authors contributed equally to this work

**Abstract** Out of the several hundred copies of *rRNA* genes arranged in the nucleolar organizing regions (NOR) of the five human acrocentric chromosomes, ~50% remain transcriptionally inactive. NOR-associated sequences and epigenetic modifications contribute to the differential expression of rRNAs. However, the mechanism(s) controlling the dosage of active versus inactive *rRNA* genes within each NOR in mammals is yet to be determined. We have discovered a family of ncRNAs, SNULs (*Single NUcleolus Localized RNA*), which form constrained sub-nucleolar territories on individual NORs and influence rRNA expression. Individual members of the SNULs monoallelically associate with specific NOR-containing chromosomes. SNULs share sequence similarity to pre-rRNA and localize in the sub-nucleolar compartment with pre-rRNA. Finally, SNULs control rRNA expression by influencing pre-rRNA sorting to the DFC compartment and pre-rRNA processing. Our study discovered a novel class of ncRNAs influencing rRNA expression by forming constrained nucleolar territories on individual NORs.

## Editor's evaluation

A long-standing question has been to understand which of the many ribosomal RNA clusters are expressed and how they are regulated. Here the authors characterize a new, noncoding RNA SNUL-1 and propose that its expression pattern in cis regulates ribosomal RNA expression.

## Introduction

The nucleolus is the most well-characterized non-membranous nuclear domain, where ribosome biogenesis and maturation occur and is formed around the nucleolus organizer regions (NORs *Németh and Grummt, 2018*). NORs are comprised of rRNA gene tandem arrays, and in human cells, they are located on the short arms (p-arm) of the five acrocentric chromosomes (Chrs. 13,14,15, 21, and 22) (*McStay, 2016*). Human cells contain >400 copies of rRNA (18 S/28 S/5.8 S) genes, yet only ~50% of the copies are transcriptionally active (*Grummt, 2007*). The expression of rRNA genes is tightly controlled during physiological processes, such as cellular development, by epigenetic mechanisms (*McStay, 2016*; *Guetg and Santoro, 2012*; *McStay and Grummt, 2008*; *Haaf et al., 1991*). However, the mechanism that precisely maintains the dosage of active versus inactive rRNA genes within a cell is yet to be determined. Two potential scenarios could explain the expression of ~50% rRNA genes at any given time. In the first scenario, rDNA repeats in half of the 10 NORs remain transcriptionally active, whereas the rRNA genes in the rest of the five NORs remain inactive (*McStay, 2023*; *Schlesinger et al., 2009*). Supporting this model, a study from Cedar laboratory reported that rRNA repeats present in the NORs of the individual alleles of Chr. 13, 14, 15, 21, and 22 replicated late during the cell cycle, and had chromatin marks consistent with transcriptionally being transcriptionally inactive. In the second scenario, each NOR contains both active and inactive rRNA clusters. In support of this claim, immune-FISH performed in several non-transformed human cell lines revealed that all the NORs with detectable rDNA repeats are active, as defined by UBF loading and their ability to form functional nucleolus (*McStay, 2023*; *van Sluis et al., 2020*). Also, recent long-read sequencing data support the hypothesis that each NOR is a mosaic of active rDNA repeat and silent rDNA repeat clusters (*van Sluis et al., 2020*; *Hori et al., 2021*). This study demonstrated that all the NORs contain methylated and unmethylated 47 S rDNA clusters, and the methylated rDNA repeats within each NOR tend to cluster together to form heterochromatic regions (*van Sluis et al., 2020*; *Hori et al., 2021*). Cis-regulatory elements and factors, including ncRNAs are suggested to maintain active versus inactive rDNA repeats within each NOR.

The nucleolus harbors a diverse set of small and long noncoding RNAs (ncRNAs), which play crucial roles in organizing the nucleolar genome as well as regulating rRNA expression (*Mamontova et al., 2021*; *Hao and Prasanth, 2022*). For example, the intergenic spacer (IGS) between rRNA genes encodes several ncRNAs, such as pRNA, PAPAS, and PNCTR, which modulate rRNA expression and nucleolus organization (*Mamontova et al., 2021*; *Hao and Prasanth, 2022*; *Bierhoff et al., 2010*; *Yap et al., 2018*). Recent studies have reported that ncRNAs, including SLERT, LoNa, AluRNAs, and the LETN lncRNAs, modulate nucleolus structure and rRNA expression via independent mechanisms (*Caudron-Herger et al., 2015*; *Wu et al., 2021*; *Li et al., 2018*; *Wang et al., 2021*). Collectively, these studies underscore the importance of ncRNAs in controlling key nucleolus functions, thus contributing to cellular homeostasis.

Besides the rDNA array, the remaining DNA sequences within the short arms of all five NOR-containing acrocentric chromosomes are highly repetitive and share high levels of sequence similarities across different chromosomes (*McStay, 2016*; *Guetg and Santoro, 2012*; *McStay and Grummt, 2008*; *Floutsakou et al., 2013*; *Németh et al., 2010*; *Németh and Längst, 2011*). As a result, insights into novel genes and/or regulatory elements within the p-arms are limited. Careful analyses of small regions located adjacent to NORs revealed that they code for lncRNAs (*Floutsakou et al., 2013*; *van Sluis et al., 2019*), indicating that the p-arms of the NOR-containing chromosomes harbor ncRNA genes, which could modulate key nucleolar functions.

In the present study, we have identified a novel family of ncRNAs: SNULs, which likely originate from the p-arms of acrocentric chromosomes and form allele-specific constrained sub-nucleolar territories on the NOR-containing chromosomes. SNUL-1 RNA displays high sequence similarity to pre-rRNA. Further studies revealed that the SNUL family of ncRNAs contributes to rRNA expression.

Thus, our study unraveled the existence of a novel family of ncRNAs that display monoallelic coating/association on the autosomal segments of NOR-containing p-arms for modulating rRNA expression.

## Results

### SNUL-1 RNA forms a distinct territory within the nucleolus

In a screen to identify cell-cycle-regulated RNAs with distinct cellular distribution (*Hao et al., 2020*), we identified a unique probe with ~600 nucleotides (*Supplementary file 1*), which hybridized to an RNA species that preferentially formed a cloud/territory within the nucleolus in a broad spectrum of human cell lines (*Figure 1A–B*; *Figure 1—figure supplement 1A–B*). Unlike other nucleolus-resident RNAs, which are homogeneously distributed in all the nucleoli within a cell, this RNA cloud decorated only one nucleolus (even in cells with several nucleoli) per nucleus in most of the diploid or near-diploid cells (*Figure 1A–C*; *Figure 1—figure supplement 1A*; embryonic stem cell [WA09], fibroblasts [WI-38, IMR-90, and MCH065], epithelial cell [hTERT-RPE-1], and lymphocyte [GM12878]). We, therefore, named the RNA *S*ingle *NU*cleolus *L*ocalized RNA-1 (SNUL-1). Strikingly, cancer cell lines displayed varied numbers of the SNUL-1 territories per nucleus (ranging from 1 to 4 SNUL-1 clouds/cell), lthough the number of SNUL-1 cloud/cell remained fixed for a particular cell line (*Figure 1B–C*; *Figure 1—figure supplement 1A*). The SNUL-1 cloud was well-preserved even in biochemically isolated nucleoli (*Figure 1D*), indicating that SNUL-1 associates with integral components of the nucleolus.

### SNUL-1 constitutes a group of RNAs with sequence features resembling 21S pre-rRNA

The original double-stranded DNA probe (Probe 1; *Figure 1—figure supplement 1C*) that detected the SNUL-1 RNA cloud(s) was mapped to hg38-Chr17: 39549507–39550130 genomic region, encoding a lncRNA. However, unique probes (non-overlapping with the probe-1 region) generated from the Chr17-encoded lncRNA failed to detect SNUL-1 RNA cloud (data not shown). Furthermore, BLAST-based analyses failed to align the Probe 1 sequence to any other genomic loci. Since a large proportion of the p-arms of nucleolus-associated NOR-containing acrocentric chromosomes is not yet annotated, we speculated that SNUL-1 could be transcribed from an unannotated genomic region from the acrocentric p-arms. RNA-FISH-based analyses revealed that a $[CT]_{20}$ repeat and a 60-nucleotide overhang sequence within the original probe-1 were crucial for detecting the SNUL-1 cloud (*Figure 1—figure supplement 1C–D*; probe 4), implying that the [AG] repeats along with unique sequence beyond the repeat contributes to the hybridization specificity and localization of SNUL-1.

During the screen, we identified another single-stranded oligonucleotide probe that shared ~73% sequence similarity to the SNUL-1 probe 4 that detected an additional RNA cloud in the nucleolus but was distinct from the SNUL-1 cloud (*Figure 1—figure supplement 1E*). We named this RNA SNUL-2. The probe that hybridized to SNUL-2 also contained an imperfect [CT]-rich region (*Figure 1—figure supplement 2A*), suggesting that both SNUL-1 and SNUL-2 RNAs contain [AG] repeats. Based on this, we propose that SNUL-1 and SNUL-2 are members of a novel RNA family and form non-overlapping constrained territories within the nucleolus.

To identify the full-length SNUL-1 sequence, we isolated total RNA from biochemically purified nucleoli (*Figure 1D*) and enriched the RNA population by performing RNA-pull down using SNUL-1 probe 4 as a bait. We then performed targeted long-read Iso-Sequencing (Iso-Seq; Single Molecule Real-Time Sequencing by PACBIO) (*Figure 1—figure supplement 2B*; please see methods for details). The binding affinities between individual high-quality full-length isoforms generated by circular consensus sequencing (CCS) reads, and SNUL-1 Probe 4 were calculated using RIBlast (see Materials and methods) (*Fukunaga et al., 2017*). Top-ranked isoforms with high binding affinity with SNUL-1 Probe 4 were picked as candidate sequences (CSs). The full-length SNUL-1 candidates identified by iso-sequencing ranged in length from 1.9 kb-3.1 kb (*Supplementary file 2*). In parallel, rRNA-depleted nucleolar RNA was also sequenced by Oxford Nanopore without SNUL-1 enrichment (*Figure 1—figure supplement 2B*). We reasoned that candidate transcripts supported by targeted PacBio sequencing and independent Nanopore long-read sequencing represent real SNUL-1 RNA candidates. Indeed, all the iso-seq SNUL-1 CSs were able to be aligned by Nanopore reads (*Figure 1—figure supplement 2C*). Aligned Nanopore reads were extracted and subjected to de novo construction of new consensus sequences (please see Materials and methods for details). Comparison between

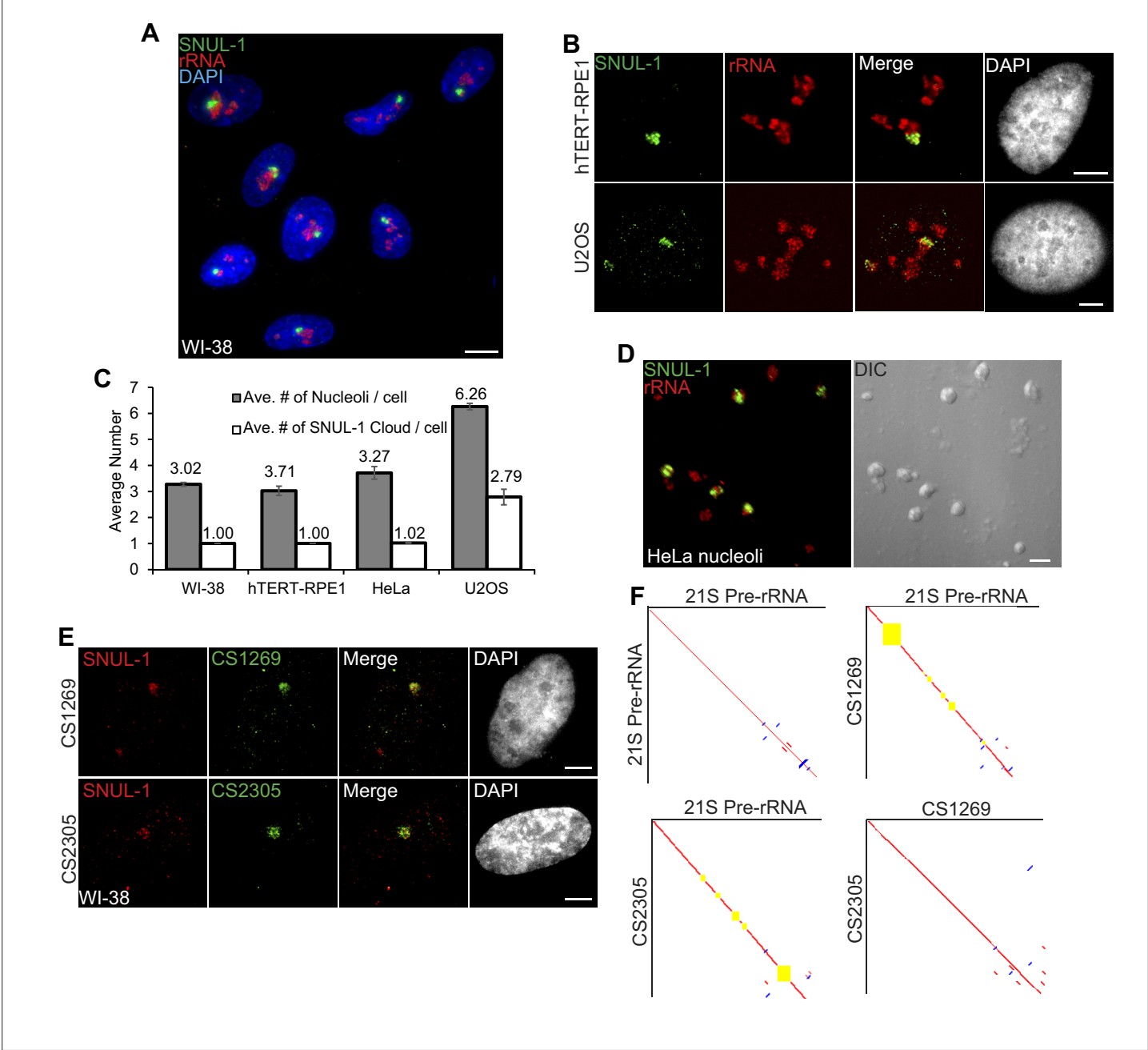

**Figure 1.** SNUL-1 forms RNA clouds in human cell lines. (**A**) RNA-FISH of SNUL-1 (green) in WI-38 cells. Nucleoli are visualized by rRNA (red). (**B**) RNA-FISH of SNUL-1 (green) in hTERT-RPE1 and U2OS cell lines. Nucleoli are visualized by rRNA (red). (**C**) Graph depicting the average number of nucleoli/cell and the SNUL-1 clouds/cell in various cell lines. n = >150 cells/experiment (**D**) RNA-FISH of SNUL-1 (green) in biochemically isolated HeLa nucleoli marked by rRNA (red). Note that the distribution of SNUL-1 is preserved in the isolated nucleoli. (**E**) RNA-FISH performed using probes designed from the SNUL-1 CSs (green) and SNUL-1 (red) Probe 1 in WI-38 cells. (**F**) Pairwise sequence comparisons between 21 S (pre-rRNA) and 21 S, 21 S and CS1269 (SNUL-1 CS), 21 S and CS2305, and CS1269 and CS2305. Red lines indicate forward aligned regions, blue lines indicate reverse aligned regions, and yellow boxes indicate unaligned regions. All scale bars, 5 μm.

The online version of this article includes the following source data and figure supplement(s) for figure 1:

**Source data 1.** Quantification of nucleoli and SNUL-1 cloud numbers in different cell lines.

**Figure supplement 1.** SNUL-1 forms RNA clouds in human cell lines.

**Figure supplement 2.** SNUL-1 forms RNA clouds in human cell lines.

**Figure supplement 3.** SNUL-1 forms RNA clouds in human cell lines.

the CSs identified by iso-seq analyses and the newly built consensus sequences by Nanopore reads revealed a ~100% identity, thus confirming these CSs in the nucleolus (*Figure 1—figure supplement 2B*). RNA-FISH was then performed using probes designed from each of the CSs. Probes designed from distinct CSs all detected RNA clouds within the same area, which were also positively detected by the original SNUL-1 cloud (*Figure 1E*; *Figure 1—figure supplement 2D*). Members of the SNUL-1 CS RNAs, though enriched with the same SNUL-1 cloud, did not display complete co-localization (*Figure 1—figure supplement 2E–F*), indicating that they might hybridize to different RNA species organized within the same territory. The full-length SNUL-1 CS RNAs identified by both iso-seq and nanopore seq. analyses contained defined 5' and 3' ends, and displayed high levels of sequence similarity between each other (>90%) (*Figure 1F*; *Supplementary file 2*). Comprehensive statistical analyses (considering the variation among the CSs and the error rate of PacBio Isoseq) rejected the hypothesis that the SNUL-1 CSs represent the same transcript with different levels of sequencing errors (*Figure 1—figure supplement 2H* & *Supplementary files 3 and 4*, see Materials and methods for the detailed analyses). Thus, it is plausible that rather than a single type of transcript, SNUL-1 represents a group of RNA species that share similar sequence features and are clustered together as an RNA cloud within the nucleolus. Pair-wise alignment of the individual members of the SNUL-1 CSs with the transcribed region of rDNA revealed ~84–91% sequence similarity between the CSs and one of the pre-rRNA species, 21 S (*Figure 1F*; *Figure 1—figure supplement 2G*). 21 S is one of the intermediates produced during rRNA processing, consisting of 18 S and partial ITS1 (*Figure 1—figure supplement 3A*). In general, the individual SNUL-1 CSs showed 84–86% alignment to the 5' end of 21 S and 81–82% alignment to the 3' end of 21 S (*Figure 1F*). For example, SNUL-1 CS2305 showed 86% identity to the first 1.3 kb of 21 S pre-rRNA, which corresponds to a significant portion of 18 S rRNA, followed by a large gap and segments of 86–91% identity corresponding to the 3' end of 18 S and the 3' end of ITS1 (*Figure 1F*, *Figure 1—figure supplement 3A*). On the other hand, CS1269 had 91% identity to the 18 S rRNA region but had a large gap in alignment in the first 1 kb, as well as segments of 84–90% identity corresponding to the 5' and 3' ends of ITS1 (*Figure 1F*).

In the nucleolus, both SNUL-1 and pre-rRNA (detected by the probe hybridizing to the internal transcribed region [ITS1] of pre-rRNA) distributed mostly non-overlapping regions, as observed by super resolution-structured illumination microscopy (SR-SIM) imaging (*Figure 1—figure supplement 3B*). In addition, modified DNA antisense oligonucleotides against SNUL (ASO-SNUL) specifically reduced only SNUL-1 levels. (*Figure 1—figure supplement 3C*). Based on these results, we conclude that SNUL-1 represents a group of novel ncRNA species showing sequence similarities to 21 S pre-rRNA and forming a single constrained territory within the nucleolus.

## RNA polymerase I-transcribed SNUL-1 is enriched at the DFC sub-nucleolar region

In the nucleolus, RNA Pol I transcription machinery is clustered in the fibrillar center (FC; marked by RNA polymerase 1 [RPA194]) and UBF (RNA pol I-specific transcription factor), allowing the transcription to happen at the outer boundary of FC (*Hozák et al., 1994*). Nascent pre-rRNAs are co-transcriptionally sorted into the dense fibrillar center (DFC; marked by fibrillarin [FBL]) located around FC for the early stages of pre-rRNA processing. The final steps of pre-rRNA processing and ribosome assembly occur in the granular component (GC; marked by B23) (*Figure 2—figure supplement 1A*). SR-SIM imaging revealed that SNUL-1 distributed across all three sub-nucleolar compartments (*Figure 2A*; *Figure 2—figure supplement 1B*) but was preferentially enriched in the DFC region (higher Pearson's correlation coefficient [PCC] between SNUL-1 and FBL [DFC marker] over RPA194 [FC marker]; *Figure 2A–B*).

The SNUL-1 cloud is also associated with the transcriptionally active DFC/FC units, as observed by the presence of 5-FU (fluro-uridine)-incorporated nascent RNA in SNUL-1-associated domains (*Figure 2C*). SNUL-1-positive regions within the nucleolus never completely overlapped with, but instead were located adjacent to nascent pre-rRNA (detected by 5'ETS-1 probe) as well as rDNA signals, as observed by SR-SIM imaging (*Figure 2D*; *Figure 2—figure supplement 1C*). Surprisingly, the areas within the nucleolus where SNUL-1 was distributed often showed weaker pre-rRNA signal (see arrowheads in *Figure 2D*). However, the SNUL-1 and nascent pre-rRNA were not colocalized but co-existed within the same FC/DFC unit, as observed and quantified by SR-SIM imaging and PCC analyses (*Figure 2E–F*). The higher PCC between two probe sets (5'ETS-1 and –2) detecting

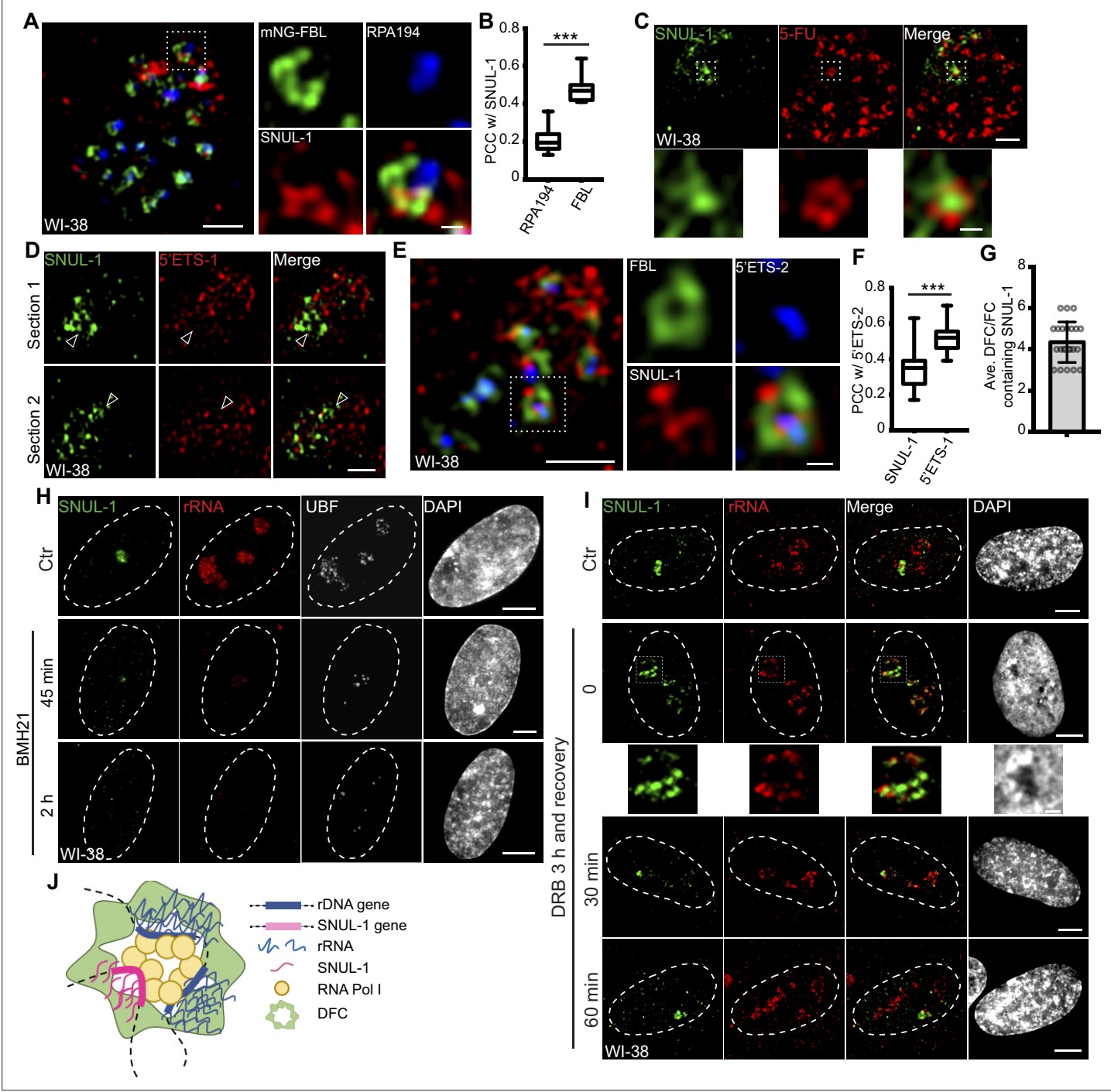

**Figure 2.** SNUL-1 is an RNA Pol I transcript and forms constrained nucleolar territory. (**A**) Representative SIM image of the SNUL-1 (red) distribution relative to DFC/FC units in WI-38 cells. FC is marked by RPA194 (blue) and DFC is marked by mNeonGreen (NG)-FBL (green). Scale bars, 1 µm (main images) and 200 nm (insets). (**B**) Box plots showing Pearson's correlation coefficients (PCCs) between SNUL-1 and either RPA194 (FC) or FBL (DFC). n=16 and 11, respectively. Statistical analysis was performed using Mann-Whitney test. *p<0.05, **p<0.01, ***p<0.001. Center line, median; box limits, upper and lower quartiles; whiskers, maximum or minimum of the data. (**C**) Representative SIM image of the SNUL-1 (green) distribution relative to the 5-FU signal (red) in WI-38 cells. Nascent RNAs are metabolically labeled by 5 min of 5-FU pulse. Scale bars, 1 µm (main images) and 200 nm (insets). (**D**) Two sections of SIM images from a single nucleolus showing relative distribution of SNUL-1 (green) and nascent pre-rRNA (marked by 5'ETS-1 probe) signals (red) in WI-38 cells. (**E**) Representative SIM image of the SNUL-1 (red) distribution relative to DFC/FC unit and pre-rRNAs in WI-38 cells. DFC is marked by FBL (green) and pre-rRNAs (blue) are detected by 5'ETS-2 probe. Scale bars, 1 µm (main images) and 200 nm (insets). (**F**) Box plots showing the Pearson's correlation coefficients (PCCs) between 5'ETS-2 signal and either SNUL-1 or 5'ETS-1 signal. n=23 and 16, respectively. Statistical analysis

*Figure 2 continued on next page*

*Figure 2 continued*

was performed using Mann-Whitney test. *p<0.05, **p<0.01, ***p<0.001. (**G**) Graph depicting the average number of SNUL-1-positive DFC/FC units/nucleolus in WI-38 cells. Center line, median; box limits, upper and lower quartiles; whiskers, maximum or minimum of the data. (**H**) Co-RNA-FISH and IF to detect SNUL-1 (green), rRNA (red) and UBF (white) in control and BMH21-treated WI-38 cells. Scale bars, 5 μm. (**I**), RNA-FISH to detect SNUL-1 (green), rRNA (red) in control and DRB-treated WI-38 cells. For recovery after DRB treatment, the drug is washed off after 3 hr of treatment and RNA-FISH is performed at 0, 30, and 60 min timepoints during recovery. Scale bars, 5 μm (main images) and 1 μm (insets). (**J**), Model showing the association of both SNUL-1 and rRNA in the same DFC/FC unit.

The online version of this article includes the following source data and figure supplement(s) for figure 2:

**Source data 1.** Quantification in *Figure 2B, F and G*.

**Figure supplement 1.** SNUL-1 is an RNA Pol I transcript and forms constrained nucleolar territory.

non-overlapping regions within the 5′ETS of pre-rRNA (*Figure 2F*; *Figure 1—figure supplement 3A*; *Figure 2—figure supplement 1D*) served as the positive control. An individual nucleolus contains dozens of FC/DFC units, with each unit containing 2–3 transcriptionally active rRNA genes *Yao et al., 2019*. SNUL-1 co-occupied with pre-rRNA in ~4 adjacent FC/DFC units within a single nucleolus (*Figure 2G*) (n=22). The localization of SNUL-1 in multiple FC/DFC units along with the observed sequence variations between SNUL-1 CSs imply that SNUL-1 RNAs are transcribed by a family of genes located in 3–4 adjacent FC/DFC units and form a single constrained sub-nucleolar cloud.

SNUL-1 is transcribed by RNA Pol I, as cells treated with RNA Pol I inhibitors (BMH21 or low dose of Actinomycin D [ActD, 10 ng/ml]) showed reduced SNUL-1 levels (*Figure 2H*; *Figure 2—figure supplement 1E*). On the other hand, the level of SNUL-1 was not significantly affected by RNA Polymerase II inhibition with drugs, including CDK9 inhibitors flavopiridol and 5,6-Dichloro-1-β-d-ribofuranosylbenzimidazole (DRB; *Figure 2I*; *Figure 2—figure supplement 1F*). However, upon RNA Pol II inhibition SNUL-1 were moved and preserved at the nucleolar periphery decorated by pre-rRNA (*Boulon et al., 2010*; *Haaf and Ward, 1996*; *Figure 2I*; *Figure 2—figure supplement 1F*; please see figure inset), and this alteration in the RNA distribution was reversible during transcription re-activation (*Figure 2I*). Together, these results indicated that SNUL-1 RNAs are transcribed by RNA Pol I in the FC/DFC region along with pre-rRNAs (*Figure 2J*).

## SNUL-1 RNA cloud associates with an NOR-containing chromosome

The nucleolus goes through disruption and reformation during mitosis (*Hernandez-Verdun, 2011*). RNA Pol I transcription is shut down from the beginning of pro-metaphase until late telophase when new nucleoli are formed (*Hernandez-Verdun et al., 2002*). During late telophase/early G1, the nucleolus is formed around the active NORs (*Savino et al., 2001*). The SNUL-1 cloud was absent from pro-metaphase to anaphase (*Figure 3—figure supplement 1A*). In the telophase/early G1 nuclei in multiple cell lines, we observed a prominent single SNUL-1 cloud co-localized with one of the several rRNAs containing active NORs (see arrow in *Figure 3—figure supplement 1A–B*). Late telophase or early G1 cells also showed a weak but distinct second SNUL-1 signal associating with another active NOR (please see arrowhead in *Figure 3—figure supplement 1A–B*). This result implies that during late/telophase early G1 nuclei, SNUL-1 decorated two independent loci.

In the interphase nuclei, only one SNUL-1 cloud was observed that was specifically associated with a single NOR-containing chromosome allele (*Figure 3A*). In this assay, the NOR-containing acrocentric chromosome arms were labeled by a probe hybridizing to the distal junction (DJ) regions, uniquely present on the p-arm of all the NOR-containing chromosomes (*Floutsakou et al., 2013*; *van Sluis et al., 2019*). This prompted us to investigate whether SNUL-1 is expressed from certain NOR(s) and if the SNUL-1 RNA is concentrated at the transcription locus on NOR(s). Further experiments revealed that in WI-38 interphase nuclei, including in G1 cells, the SNUL-1 cloud is specifically associated (100% association) with one allele of Chr. 15 (*Figure 3B–C*). This was demonstrated by co-RNA and DNA-FISH, which detected SNUL-1 cloud and Chr. 15 markers, including Chr. 15 q-arm paint (*Figure 3B–C*; *Figure 3—figure supplement 1I*), Chr. 15-specific centromere (15CEN; α-Satellite or 15p11.1-q11.1), and peri-centromeric Satellite III repeats (15Sat III repeats or 15p11.2) (*Figure 4A*). The association rate between the SNUL-1 cloud and other NOR-containing chromosomes, such as Chr. 13 and Chr. 22 was 30–40%, which could be because these chromosomes are localized in the same nucleolus, along with the SNUL-1-associated Chr. 15 allele. Monoallelic association of SNUL-1 to Chr. 15 was also confirmed in several other human cell lines, including in primary fibroblasts (IMR-90 [lung], MCH065

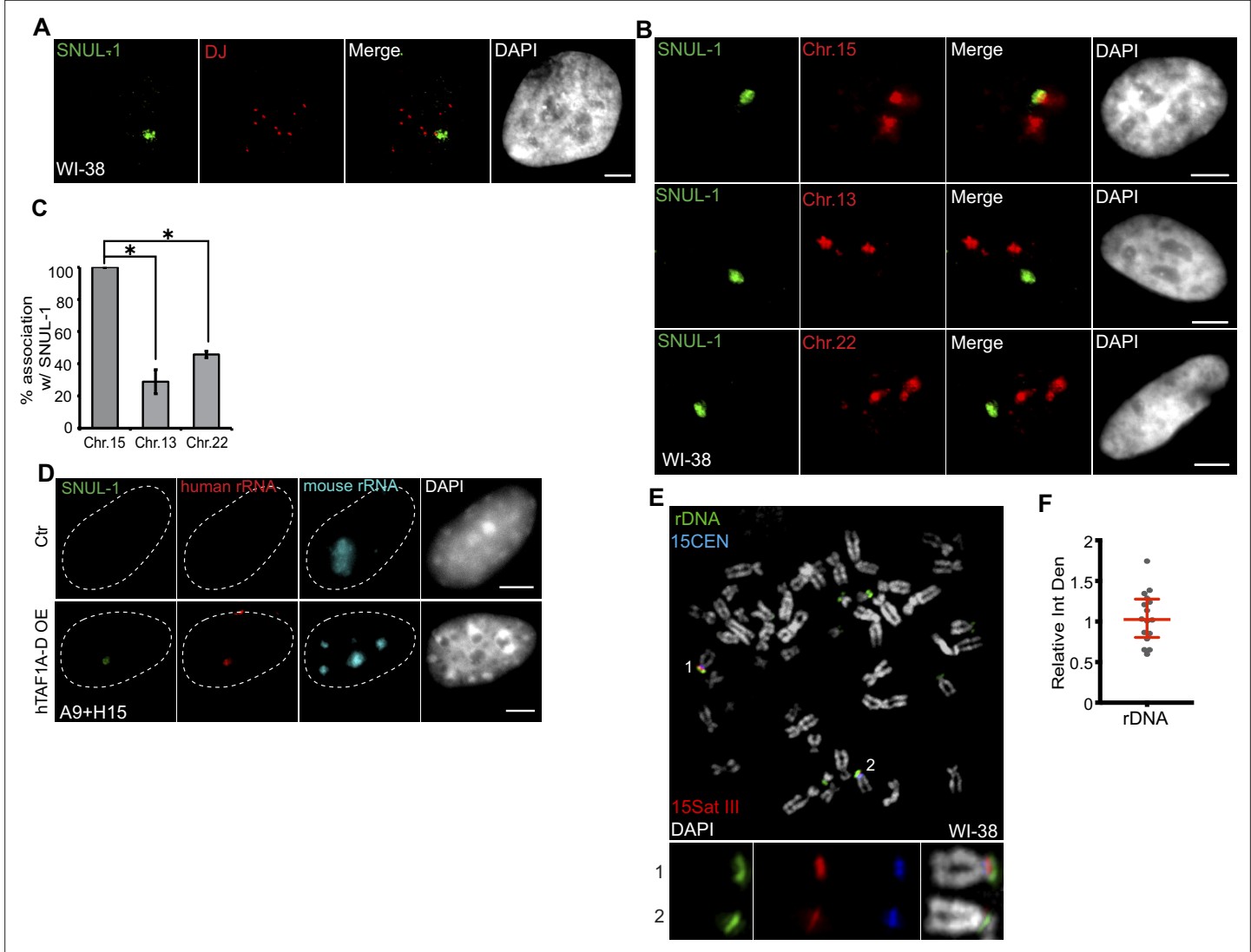

**Figure 3.** SNUL-1 is associated with the NOR of one Chr. 15 allele. (**A**) DNA-RNA-FISH of SNUL-1 RNA (green) and distal junction (DJ) DNA (red) in WI-38 cells. (**B**) DNA-RNA-FISH of SNUL-1 RNA and Chr. 15, Chr. 13, and Chr. 22 marked by probes painting the q-arms of the chromosomes in WI-38 cells. (**C**) Quantification of the association rates between SNUL-1 and NOR containing chromosomes. Data are presented as Mean ± SD from biological triplicates.>50 cells were counted for each of the biological repeats. Student's unpaired two-tailed t-tests were performed. *$P<0.05$. (**D**) RNA-FISH to detect SNUL-1 (green), human rRNA (red) and mouse rRNA (blue) in control and hTAF1A-D overexpressed A9 +H15 cells. Dotted lines mark the boundary of the nuclei. (**E**) DNA-FISH showing rDNA and CEN15 and 15 Sat III contents on Chr. 15 in WI-38 metaphase chromosomes. The two alleles of Chr.15 are marked by 15Sat III and 15CEN, rDNA arrays are detected by a probe within the IGS region (See *Figure 1—figure supplement 3A*). (**F**) Relative integrated density of the two Chr. 15 rDNA arrays is calculated by dividing the measurement of the rDNA signal on the Chr. 15 with larger 15Sat III by that of the one on the Chr.15 with smaller 15Sat III. All scale bars, 5 µm. DNA is counterstained with DAPI.

The online version of this article includes the following source data and figure supplement(s) for figure 3:

**Source data 1.** Quantification of association rates between SNULs and different chromosomes in *Figure 3C*.

**Figure supplement 1.** SNUL-1 is associated with the NOR of one Chr. 15 allele.

**Figure supplement 1—source data 1.** Quantification of association rates between SNULs and different chromosomes in *Figure 3—figure supplement 1F*.

**Figure supplement 1—source data 2.** Quantification of relative integrated density of the rDNA on the two Chr.15 alleles in *Figure 3—figure supplement 1H*.

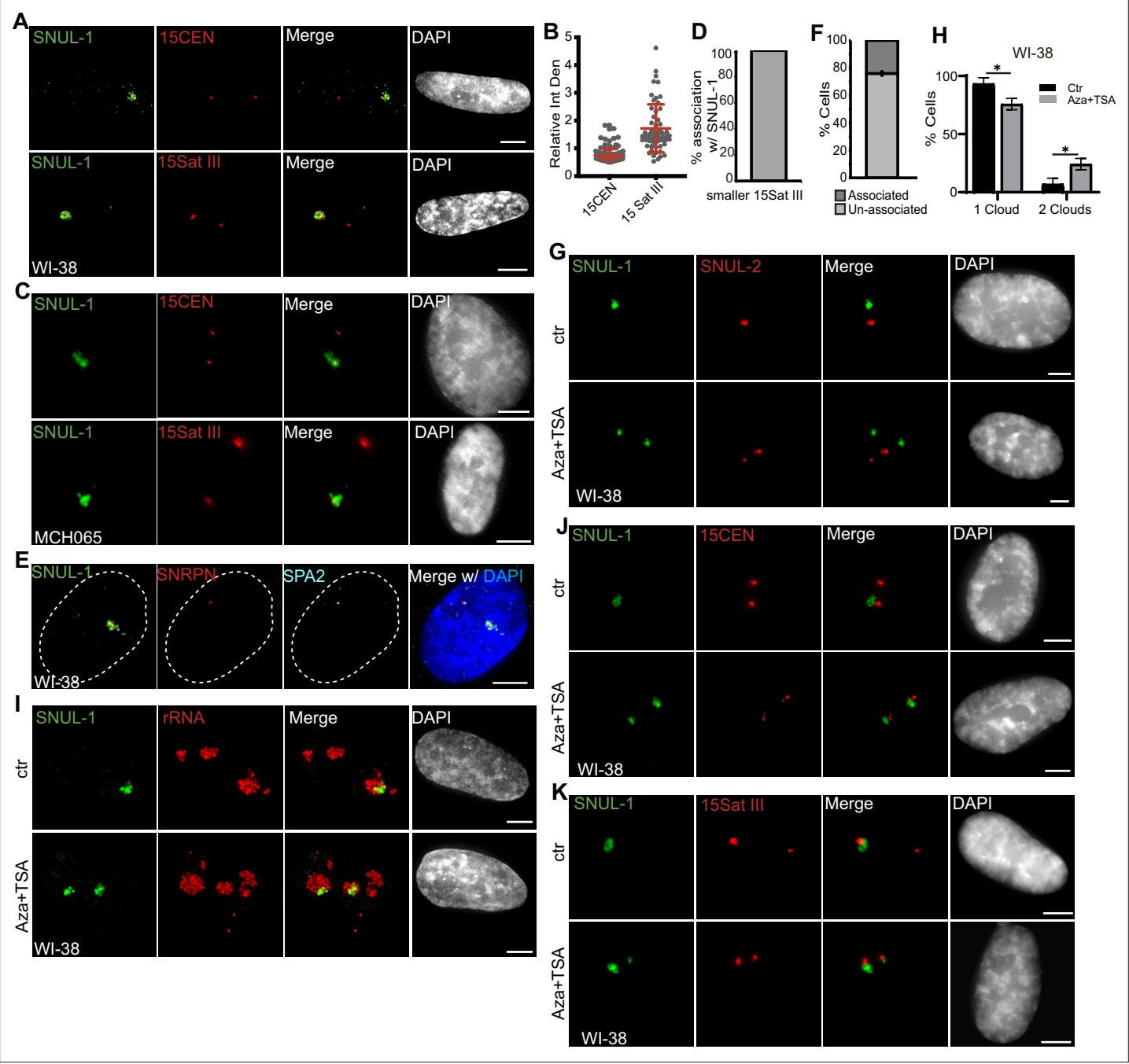

**Figure 4.** The SNUL-1 cloud displays mitotically-inherited random monoallelic association. (**A**) Representative RNA-FISH images showing the association of SNUL-1 with the 15Sat III or 15CEN in WI-38 nuclei. (**B**) Plot showing the relative integrated density of the 15Sat III signal in WI-38 nuclei. The relative integrated density is calculated by dividing the measurement of the larger DNA-FISH signal by that of the smaller DNA-FISH signal. Data are presented as Median and interquartile range. n=60. (**C**) Representative RNA-FISH images showing the association of SNUL-1 cloud with 15Sat III or 15CEN in MCH065 nuclei. (**D**) Quantification of the association rates between SNUL-1 and the smaller 15Sat III in MCH065 nuclei. Data are presented as Mean ± SD from biological triplicates.>50 cells were counted for each of the biological repeats. (**E**) Representative RNA-FISH images showing the localization SNUL-1 along with the SNRPN and SPA2 transcription site on the paternal allele of Chr. 15 in WI-38 nucleus. Dotted lines mark the boundary of the nucleus. (**F**) Quantification of the association rate between SNUL-1 and the transcription sites of SNRPN and SPA2 in WI-38 cells. Data are presented as Mean ± SD from biological triplicates.>35 cells were counted for each of the biological repeats. (**G**) Representative RNA-FISH images showing the distribution of SNUL-1 (green) and SNUL-2 (red) in control and Aza-dC (500 nM) and TSA (80 nM) treated WI-38 nuclei. (**H**) Quantification of the percentage of cells showing one or two SNUL-1clouds in control and Aza +TSA-treated WI-38 cells. Data are presented as Mean ± SD from biological triplicates.>50 cells were counted for each of the biological repeats. Student's unpaired two-tailed t-tests were performed. *p<0.05. (**I**), RNA-

*Figure 4 continued on next page*

*Figure 4 continued*

FISH to detect SNUL-1 clouds in control and Aza +TSA-treated WI-38 nuclei. Nucleoli are visualized by rRNA (red). (**J**) DNA-RNA-FISH of SNUL-1 RNA and 15CEN in control and Aza +TSA-treated WI-38 nuclei. (**K**) DNA-RNA-FISH to detect SNUL-1 RNA and 15Sat III in control and Aza +TSA-treated in WI-38 nuclei. All scale bars, 5 µm. DNA is counterstained by DAPI.

The online version of this article includes the following source data and figure supplement(s) for figure 4:

**Source data 1.** Quantification in *Figure 4D, F and H*.

**Figure supplement 1.** The SNUL-1 cloud displays mitotically-inherited random monoallelic association.

**Figure supplement 1—source data 1.** Quantification in *Figure 4—figure supplement 1B,D,H and J*.

[dermal]), hTERT-immortalized near-diploid retinal pigment epithelial cells (hTERT-RPE1) and Fibrosarcoma (HTD114) cells (*Figure 3—figure supplement 1C–D*; *Figure 4C*; *Breger et al., 2004*; *Gupta et al., 1997*). Like SNUL-1, SNUL-2 RNA cloud is specifically associated with one allele of the NOR-containing Chr.13 (*Figure 3—figure supplement 1E–F*). Based on these results, we hypothesize that a unique subset of SNUL-like genes may be present in each of the five NOR-containing acrocentric chromosome arms, where each member of the SNUL RNA (SNUL-1 and SNUL-2 on Chr.15 and 13, respectively) form spatially constrained RNA territories on the p-arm of the chromosome alleles.

By utilizing the mouse A9 cells integrated with one allele of human Chr. 15 (mono-chromosomal somatic cell hybrid A9 +H15) (*van Sluis et al., 2019*), we further confirmed that SNUL-1 is indeed transcribed from Chr. 15 by RNA pol I and formed a confined RNA territory in the nucleolus. In the somatic-hybrid cells, the NOR on the transferred human chromosome remained silenced and showed no human rRNA expression due to the inability of mouse-encoded RNA Pol I-specific transcription factors to bind to the human RNA pol I-transcribed gene promoters (*Figure 3D*; Ctr). Exogenous expression of human TBP-associated factors (TAF1A-D) in the A9 +H15 cells reactivated RNA Pol I transcription from human Chr. 15, reflected by the presence of both human rRNA and SNUL-1 in the nucleolus (*Figure 3D*; *van Sluis et al., 2019*; *Murano et al., 2014*).

The rDNA content between the two alleles could vary profoundly in cell lines, as recently reported in the case of hTERT-RPE1 (*van Sluis et al., 2020*; see also *Figure 3—figure supplement 1G–H*). Quantification of the integrated density of the rDNA spots on the mitotic chromosome spreads of WI-38 confirmed equal rDNA content between the two Chr.15 alleles (*Figure 3E–F*; green). This indicates that the monoallelic association of SNUL-1 to Chr. 15 in WI-38 cells is not dictated by the rDNA content in these cells. We further observed that SNUL-1-decorated Chr. 15 allele contained transcriptionally active rDNA clusters as shown by positive 5-FU incorporation (*Figure 3—figure supplement 1J*). Previous studies have established RNA pol I transcription factor, UBF as a marker for transcriptionally active NORs (*McStay, 2023*; *van Sluis et al., 2020*; *van Sluis et al., 2016*). We observed the presence of UBF in both the Chr. 15 alleles, implying that SNUL-1-decorated NORs contain transcriptionally active rDNA clusters (*Figure 3—figure supplement 1I*).

## SNUL-1 RNA displays mitotically inherited random monoallelic association (rMA) to the NOR of Chr. 15

We consistently observed a significant difference in the size of the Chr. 15-specific peri-centromeric Sat III repeat (15Sat III) signal between the two Chr. 15 alleles in several of the diploid or pseudo-diploid cell lines ([WI-38; *Figure 3E*; *Figure 4A–B*], [hTERT-RPE-1; *Figure 3—figure supplement 1G* & *Figure 4—figure supplement 1A–B*], [MCH065; *Figure 4C*]), implying that these cells showed allele-specific differences in the amount or compaction of peri-centromeric 15Sat III DNA. Interestingly, in 100% of WI-38 and hTERT-RPE1 cells, the SNUL-1 cloud was associated only with the larger 15Sat III signal containing Chr.15 allele (*Figure 4A–B*; *Figure 4—figure supplement 1A–B*; n=50 from biological triplicates). On the other hand, the SNUL-1 cloud in the MCH065 cells was associated with the Chr. 15 allele containing the smaller 15Sat III signal (*Figure 4C–D*). These results suggest that SNUL-1 is either imprinted or display mitotically inherited random monoallelic association in a cell type-specific manner. Loss-of-function studies revealed that SNULs did not influence allele-specific 15Sat III levels or compaction (*Figure 4—figure supplement 1C–D*).

We next determined whether SNUL-1 non-randomly associates with the paternal or maternal allele of the Chr. 15. Genes encoded within the imprinted Prader-Willi Syndrome (PWS)/Angelman Syndrome (AS) genomic loci (*Nicholls and Knepper, 2001*), such as *SNRPN* and the lncRNA *SPA2*,

are expressed only from the paternal allele of Chr. 15 (*Wu et al., 2016*; *Morcos et al., 2011*). In WI-38 cells (n=75), the SNUL-1 cloud was preferentially located away from the paternal Chr.15 allele, co-expressing *SNRPN* and *SPA2* (*Figure 4E–F*), indicating that in WI-38 cells, SNUL-1 associated with the maternal allele of Chr. 15, which also contains larger 15SatIII. On the other hand, in the MCH065 Fibroblasts, MCH065-derived iPSCs (MCH2-10), and HTD114 cells, the SNUL-1 cloud was associated with the paternal Chr. 15 allele, as demonstrated by the localization of SNUL-1 cloud next to the smaller 15SatIII or *SNRPN* RNA signals (*Figure 4C–D*; *Figure 4—figure supplement 1E–M*). In WI-38, hTERT-RPE1, and MCH2-10, the smaller 15SatIII signal was always associated with the paternal Chr. 15 (*Figure 4—figure supplement 1I–J*). Based on these results, we conclude that SNUL-1 is not an imprinted gene but rather displays mitotically inherited random monoallelic association (rMA) to paternal or maternal Chr. 15 in a cell line-specific manner (*Figure 4—figure supplement 1M*; *Reinius and Sandberg, 2015*; *Chess, 2016*).

Repressive epigenetic modifiers control imprinted or monoallelic expression of lncRNAs (*Yang et al., 2003*; *Loda et al., 2022*; *Lessing et al., 2013*). WI-38 cells co-incubated with 5-Aza-2'-deoxycytidine (5-Aza-dC), which inhibits DNA methyl transferase (DNMT) and Trichostatin A (TSA), which inhibits histone deacetylase (HDAC) showed two separate SNUL-1 and SNUL-2 foci (*Figure 4G–H*). Both the SNUL-1 clouds in 5-Aza-dC+TSA-treated cells remained localized in the nucleolus (*Figure 4I*) and were associated with both Chr. 15 alleles in a population of cells (*Figure 4J–K*). These results indicate the potential involvement of repressive epigenetic regulators in maintaining the monoallelic association of SNULs.

## SNUL RNAs influence rRNA biogenesis

We next evaluated the potential involvement of SNUL territory in nucleolar functions. Our Iso-Seq and imaging data revealed that *SNUL-1* constituted a family of genes/transcripts sharing high sequence similarity. Modified antisense oligonucleotides (ASOs; ASO112 and ASO113) targeting individual SNUL-1 CS candidates did not reduce total SNUL-1 levels (*Figure 5—figure supplement 1A*). The repeat sequence within SNULs was highly conserved among all the SNUL-1 CSs and was also shared by the SNUL-2 transcript. Using an ASO targeting this region (ASO-SNUL), we efficiently depleted both SNUL-1 and SNUL-2 (*Figure 5A*, *Figure 5—figure supplement 1A*). Interestingly, SNUL-depleted WI-38 showed enhanced 5-FU incorporation in the nucleolus (*Figure 5B–C*) and showed increased levels of nascent 47 S pre-rRNA, quantified by single molecule RNA-Fluorescent hybridization (smRNA-FISH) using a probe set (5'ETS-2) that preferentially detects nascent 47 S pre-rRNA (*Figure 5—figure supplement 1B–C*). A similar increase in the pre-rRNA levels was also observed upon SNUL depletion in other cell lines, including RPE-1 (SNUL-1 associates with the maternal Chr.15 allele) and HTD114 (SNUL-1 associates with the paternal Chr.15 allele; *Figure 5—figure supplement 1D–G*). In addition, we quantified the pre-rRNA levels in a population of RPE1 cells using the recently developed RNA FISH-FLOW (*Antony et al., 2022*) assay and observed a similar increase in pre-rRNA levels upon SNUL depletion (*Figure 5—figure supplement 1H*). Both SNUL-ASO and the SNUL-1 RNA-FISH probe contain the CT-repeat. The SNUL-ASO may interfere with RNA-FISH hybridization by competing for the similar target sequence within the SNUL-1 RNA. To test the competition model (ASO and SNUL-1 probe competing for the SNUL-1 RNA), we fixed control and SNUL-ASO-treated RPE1 cells, denatured the cells, and performed highly stringent washes to strip the ASOs from the target RNA. After that, we performed RNA-FISH to detect SNUL-1. Results revealed loss of SNUL-1 signal in the cells treated with SNUL-1 ASO even post-denaturation (*Figure 5—figure supplement 1I*). If the lack of SNUL-1 signals in SNUL ASO-treated cells is because the SNUL ASO compete with the probe for SNUL-1 RNA is true, then upon denaturation, the target SNUL-1 RNA, disassociated from the SNUL ASO would have shown positive RNA signal with SNUL-1 probe. Based on this observation, we infer that even though the competition could still be possible, it cannot explain the complete loss of SNUL-1 RNA signal in the ASO-treated samples. Our results imply that SNUL depletion either enhanced the expression of nucleolus-enriched *rRNA* genes and/or altered the co-transcriptional pre-rRNA processing.

We observed that SNUL-depleted cells showed an increased number of FC/DFC compartments/nucleolus (*Figure 5D and F*; *Figure 5—figure supplement 2A–B*) by performing SIM imaging of nucleolar proteins decorating specific sub-nucleolar compartments. Earlier studies have documented the roles of nucleolus-enriched lncRNAs in nucleolus functions by modulating the overall organization

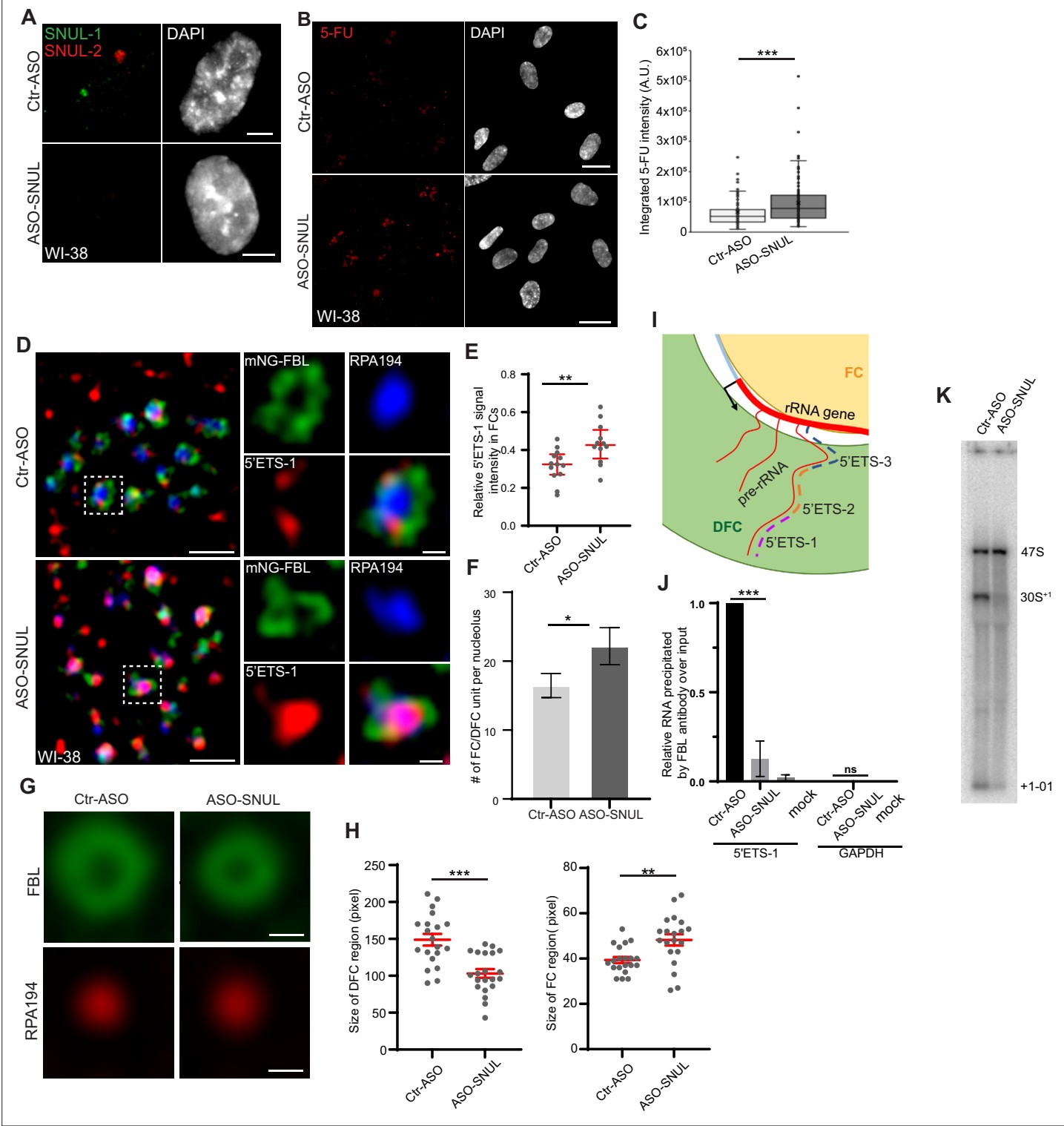

**Figure 5.** SNUL-1 influences rRNA biogenesis. (**A**) RNA-FISH of SNUL-1 (green) and SNUL-2 (red) in WI-38 cells transfected with ctr-ASO or ASO-SNUL oligonucleotides. (**B**) 5-FU immunostaining in control and SNUL-depleted WI-38 cells. Scale bars, 5 μm. Ctr-ASO and ASO-SNUL-treated cells were pulse labeled by 5-FU for 20 min. Scale bars, 20μm. (**C**) Boxplots of integrated 5-FU intensity per nucleus in control and SNUL-depleted WI-38 cells. Center line, median; box limits, upper and lower quartiles; whiskers, maximum or minimum of the data. Mann-Whitney test is performed. n=100. *p<0.05, **p<0.01, ***p<0.001. (**D**), SIM image of a single nucleolus showing the nascent pre-rRNA detected by 5'ETS-1 probe (red) in Ctr and SNUL-depleted cells. DFC is marked by mNG-FBL (green) and FC is marked by RPA194 (blue). Scale bars, 1 μm (main images) and 200 nm (insets). (**E**), Quantification of relative 5'ETS-1 intensity in FC of nucleolus in Ctr-ASO and ASO-SNUL cells. Center line, median. Mann-Whitney test is performed. **p<0.01.>50

*Figure 5 continued on next page*

*Figure 5 continued*

DFC/FC units from 10 to 15 nucleoli were counted for each sample. (**F**) Quantification of the average number of FC/DFC unit per nucleolus in control and SNUL-depleted WI-38 cells. Data are presented as Mean ± SEM from nine biological repeats.>15 nucleoli were counted from each experiment. *p<0.05. (**G**) Averaged images of FC or DFC unit in control and SNUL-depleted WI38 cells. DFC is marked by FBL and FC is marked by RPA194. Scale bars, 200 nm. (**H**) Quantification of FC and DFC unit sizes in control and SNUL-depleted WI-38 cells. Data are presented as Mean ± SEM. Center line, median. Mann-Whitney test is performed. **p<0.01, ***p<0.001. 20 DFC/FC units from different cells were counted. (**I**) Schematic showing the sorting of pre-rRNA in DFC/FC unit. The position of the 5′ETS regions are marked. (**J**) Relative 5′ETS-1 precipitated by FBL antibody in control and SNUL-depleted WI-38 cells, ***p<0.001. (**K**) Northern blot using 5-ETS-1 probe from total RNA isolated from control and SNUL-depleted WI-38 cells.

The online version of this article includes the following source data and figure supplement(s) for figure 5:

**Source data 1.** Quantification in *Figure 5C, E, F and H*.

**Source data 2.** Uncropped image of Northern Blot in *Figure 5J*.

**Figure supplement 1.** SNUL-1 influences rRNA biogenesis.

**Figure supplement 1—source data 1.** Quantification of pre-rRNA signal intensities in different cell lines, related to *Figure 5—figure supplement 1C,E,G*.

**Figure supplement 2.** SNUL-1 influences rRNA biogenesis.

**Figure supplement 2—source data 1.** Uncropped image of Northern Blot in 1511 *Figure 5—figure supplement 2C and D*.

**Figure supplement 3.** SNULs influence Chr.15 replication timing.

**Figure supplement 3—source data 1.** Quantification of ReTiSH assay showing the relative integrated density of the rDNA signal on the two Chr.15 alleles, related to *Figure 5—figure supplement 3C*.

**Figure supplement 4.** SNULs influence Chr.15 replication timing.

**Figure supplement 5.** SNULs influence Chr.15 replication timing.

**Figure supplement 6.** rDNA clusters do not show allele-specific histone modifications.

**Figure supplement 6—source data 1.** Quantification of histone mark signal intensities on Chr.15 alleles in HTD114 cell line, related to *Figure 5—figure supplement 6D*.

**Figure supplement 7.** rDNA clusters do not show allele-specific histone modifications.

**Figure supplement 7—source data 1.** Quantification of histone mark signal intensities on Chr.15 alleles in RPE1 cell line, related to *Figure 5—figure supplement 7D*.

of sub-nucleolar compartments (*Caudron-Herger et al., 2015*; *Wu et al., 2021*; *Xing et al., 2017*). SNUL-depleted cells specifically showed smaller fibrillarin-stained DFC rings and larger RPA194 decorated FC domains, as observed in averaged images of FC and DFC domains using Average Intensity Projections (n=20; *Figure 5G–H*). The changes in the number and size of FC/DFC domains in SNUL-depleted cells could be a consequence of altered pre-rRNA expression in these cells.

The nascent 47 S pre-rRNA is co-transcriptionally sorted from its transcription site at the DFC/FC boundary to DFC (*Yao et al., 2019*). The DFC-localized FBL binds to the 5′ end upstream of the first cleavage site (01 site; *Figure 1—figure supplement 3A*; *Figure 5I*) of the 47 S pre-rRNA co-transcriptionally and facilitates pre-rRNA sorting for efficient RNA processing and DFC assembly (*Yao et al., 2019*). Due to this, the 5′ end of 47 S pre-rRNA is localized in the DFC region, as shown by SR-SIM of smRNA-FISH using the 5′ external transcribed spacer (5′ETS)–1 probe set targeting the first 414 nts of 47 S pre-rRNA (*Figure 5D*; *Yao et al., 2019*). On the other hand, the region within the 47 S pre-rRNA 5′ETS located downstream of the 01 cleavage site (detected using 5′ETS-2 & 3 probes [*Figure 1—figure supplement 3A*; *Figure 5I*; probes: 5′ETS-2 and 3]) associated with the rRNA transcription sites (FC or FC/DFC junction) (*Figure 5—figure supplement 2A–B*; *Yao et al., 2019*). Interestingly, in the SNUL-depleted cells, the FBL-interacting 5′ETS-1 region within the 47 S pre-rRNA failed to sort to DFC and instead preferentially accumulated in the FC (*Figure 5D–E*). An earlier study demonstrated that depletion of pre-rRNA processing factors, such as FBL, compromised 47 S pre-rRNA sorting at DFC, resulting in the accumulation of pre-rRNA in the FC region (*Yao et al., 2019*). Our results suggest the possibility that SNULs could influence pre-rRNA biogenesis by modulating FBL-mediated pre-rRNA sorting/processing. In support of this, we observed an enriched association of SNUL-1 in the FBL-localized DFC (*Figure 2A–B*). We, therefore, determined whether SNULs influence the interaction between FBL and pre-rRNA in DFC. Toward this, we performed FBL RNA-immunoprecipitation followed by quantitative RT-PCR to quantify the interaction between FBL and nascent pre-rRNA in control and SNUL-depleted cells. Strikingly, SNUL-depleted cells showed

reduced interaction between FBL and pre-rRNA (*Figure 5J*), indicating that DFC-enriched SNULs could influence the FBL interaction with pre-rRNA.

The sequence upstream of the 01-cleavage site within the 47 S pre-rRNA (detected by 5'ETS-1 probe), is co-transcriptionally cleaved after it is sorted to DFC by FBL. Defects in the pre-rRNA sorting to DFC were shown to affect pre-rRNA processing (*Yao et al., 2019*). SNUL-1-depleted cells showed defects in the initial cleavage at the 5'end of the 47 S pre-rRNA, as observed by the reduced levels of 30S+1 intermediate and +1–01 cleaved product (*Figure 5K*; *Figure 5—figure supplement 2C–D*) by Northern blot analyses. The +1–01 is the unstable product processed from the 5'end of 47 S pre-rRNA due to the cleavage at the 01 site. These results indicate the potential involvement of SNULs in pre-rRNA, sorting and/or co-transcriptional processing.

NcRNAs influence asynchronous DNA replication in a monoallelic fashion. For example, Thayer laboratory has identified a group of lncRNAs, ASARs (Asynchronous replication and Autosomal RNAs), which are monoallelically transcribed and associated on various chromosomes and control asynchronous replication timing between chromosome alleles (*Heskett et al., 2020*; *Heskett et al., 2022*; *Donley et al., 2015*). An earlier study from Cedar laboratory reported allelic-specific replication and transcription of rDNA loci in human cells (*Schlesinger et al., 2009*). We, therefore, tested whether SNULs contribute to asynchronous replication and monoallelic expression of rDNA loci in human cells. We performed the ReTiSH (Replication timing-specific hybridization), an assay previously developed by Cedar laboratory, to determine the replication timing of both the alleles of Chr. 15 in control and SNUL-depleted HTD114 and WI-38 cells (please see Materials and methods for details about ReTisH; *Schlesinger et al., 2009*; *Donley et al., 2015*). Briefly, control and SNUL-depleted cells were incubated with BrdU for specified time points (6 or 14 hr) and then synchronized in metaphase by treating the cells with colcemid for 1 hr. In these cells, 6 hr BrdU incubation, cells present in late S-phase preferentially incorporates BrdU, whereas 14 hr, labels cells in any stage of S phase. The replication status of rDNA loci was determined by a modified version of co-FISH, an assay in which the BrdU +ve replicated regions are specifically converted to ssDNA and then hybridized with rDNA probes (*Cornforth and Eberle, 2001*). In addition to rDNA, we also labeled the mitotic chromosomes with a CEN15 probe to mark the Chr.15 rDNA clusters. In contrary to what had been reported by the Cedar group, we did not observe allele-specific differential replication of rDNA clusters on the Chr.15 p-arm. Both the alleles of Chr.15 p-arms replicate during the late S phase in both the cell lines (please see Ctrl in *Figure 5—figure supplement 3A–B*; *Figure 5—figure supplement 4A–B*). Surprisingly, upon SNUL-depletion, the Chr. 15 alleles showed differential replication timing. In SNUL-depleted HTD114 cells, the maternal allele of Chr.15 p-arm (allele with larger CEN15 signal) replicated earlier (no rDNA signal in 6 hr) compared to the paternal allele, which continued to replicate during late S-phase (*Figure 5—figure supplement 3A,C*; compare 6 hr Ctrl versus 6 hr ASO-SNUL).

To independently test the ReTiSH results, we performed EdU-incorporation assays in control and SNUL-depleted interphase HTD114 cells. Cells were pulse-labelled with EdU to mark replication sites (green), co-stained with rDNA (Cyan) and Chr.15-specific centromeres (yellow) to determine the replication timing of the Chr.15 p-arm containing rDNA cluster (*Figure 5—figure supplement 5A–C*). In control cells, rDNA cluster from both the chr-15 p-arms co-localized with EdU +ve foci predominantly in mid and late S-phase cells. Upon SNUL depletion, Chr. 15 rDNA cluster from the maternal allele showed enhanced association with EdU foci in the middle S-phase cells (*Figure 5—figure supplement 5A–C*). These results implicate that the replication timing of maternal allele shifts from late S-phase to middle S-phase upon SNUL depletion, confirming the ReTiSH results. However, the significance of this observation remains to be elucidated.

To test whether SNULs influence monoallelic transcription of rDNA clusters, we evaluated the status of monoallelic association of three of the active histone marks (H3Ac, H4Ac, H3K4me3) on the Chr.15 p-arms in the mitotic chromosome spreads of control and SNUL-depleted HTD114 and RPE1 cells, based on the assay previously described in the Cedar study. We observed active histone mark signals in all the chromosome arms, including the q-arms of chr. 15 (*Figure 5—figure supplement 6*; *Figure 5—figure supplement 7*; *Schlesinger et al., 2009*). However, unlike what had been reported earlier by the Cedar lab in their earlier study, we could not find any active histone marks in both the alleles of rDNA bearing p-arms of Chr. 15 (*Figure 5—figure supplement 6*; *Figure 5—figure supplement 7*) in control as well as SNUL-depleted cells. We measured the histone mark signal intensities between two chr. 15 p-arm alleles and did not find any significant difference between the two

alleles in control and SNUL-depleted cells. In conclusion, we did not find monoallelic replication and transcription of chr. 15 rDNA cluster containing NORs in human cell lines (hTERT-RPE1, HTD114, and WI-38) those display monoallelic association of SNULs, supporting the notion that every NORs contain both active and inactive rDNA clusters. Based on these results, we argue that SNUL RNAs constitute the first family of ncRNAs in the nucleolus, which displays monollelic association to specific NOR-containing chromosomes.

## Discussion

We have discovered SNULs, a novel family of ncRNAs, which display non-random association to specific NOR-containing chromosomes within the nucleolus. Our data suggest that SNUL-1 is a member of a family of RNAs sharing similar sequence features. The most striking feature of the SNUL-1 sequence is its resemblance to the 21 S pre-rRNA intermediate. Recent genomic mapping of acrocentric chromosome arms revealed that most of the sequences in the NOR-containing p-arms are shared among all 5 chromosomes (*Floutsakou et al., 2013*; *van Sluis et al., 2019*; *Nurk et al., 2022*). However, these studies have also identified inter-chromosomal sequence variations (*van Sluis et al., 2019*). Our observations showing the association of individual members of SNULs to specific alleles of one of the acrocentric chromosomes support the idea that the acrocentric arms encode chromosome- and allele-specific transcripts.

The underlying mechanism(s) controlling differential expression of rRNA gene copies in mammals is yet to be determined. Nucleolar dominance (NuD) is a developmentally regulated process that is speculated to act as a dosage-control system to adjust the number of actively transcribed rRNA genes according to the cellular need (*McStay and Grummt, 2008*; *Grummt and Pikaard, 2003*). However, NuD is primarily observed in the 'interspecies hybrids' of plants, invertebrates, amphibians, and mammals (*Tucker et al., 2010*; *McStay, 2006*; *Cassidy and Blackler, 1974*; *Goodrich-Young and Krider, 1989*; *Croce et al., 1977*). NuD is reported in certain nonhybrid or 'pure species' of plants and fruit flies but has not yet been observed in mammals, primarily due to a lack of information about the allele-specific rRNA sequence variations (*Lewis et al., 2004*; *Greil and Ahmad, 2012*; *Xie et al., 2012*). During NuD, chromosome allele-specific rRNA expression is observed, in which the rRNA gene array within the NOR that is inherited from one parent (dominant) is maintained in a transcriptionally active status, while the rRNA loci from the other parent (under dominant) are preferentially silenced (*McStay, 2006*; *Pontes et al., 2007*; *Earley et al., 2010*; *Chen et al., 1998*). Several features associated with the monoallelic regulation of SNULs share similarities with NuD (*Tucker et al., 2010*; *McStay, 2006*). SNUL-1 and rRNA expression in NuD is dictated by non-imprinted random monoallelic expression. Repressive epigenetic modifiers play vital roles in the allele-specific expression of SNULs, and rRNAs during NuD (*Earley et al., 2010*; *Chen and Pikaard, 1997*; *Neves et al., 1995*). Interestingly, specific sequence elements located near the NORs are implicated in controlling NuD (*Labhart and Reeder, 1984*; *Rabanal et al., 2017*; *Warsinger-Pepe et al., 2020*). For example, in the germ cells of male *Drosophila*, Y chromosome sequence elements promote developmentally regulated NuD of the Y chromosome-encoded rRNA over the X-chromosome rRNA (*Warsinger-Pepe et al., 2020*).

We observe that SNUL-1 forms a spatially constrained RNA territory that associates next to the NOR of Chr. 15 but is devoid of pre-rRNA. Furthermore, SNUL-depleted cells show elevated levels of pre-rRNA, along with defects in pre-rRNA sorting and processing. One-way SNUL-1 could modulate rRNA expression is via modulating the levels of bioprocessing machinery that control rRNA biogenesis and processing. For example, high sequence similarity between SNUL-1 and pre-rRNAs helps SNUL-1 to compete for and/or recruit factors that regulate rRNA biogenesis in a spatially constrained area within the nucleolus. A recent study, by visualizing the distribution of tagged pre-rRNAs from an NOR-containing chromosome, reported that similar to SNULs, pre-rRNAs transcribed from individual NORs form constrained territories that are tethered to the NOR-containing chromosomal regions (*Mangan and McStay, 2021*). It is possible that SNUL-1, by forming a distinct RNA territory on the NOR of the Chr. 15 allele, influences the expression of rRNA genes in cis. Such organization of SNULs and rRNA territories in a constrained area within the nucleolus would help to control the expression of a subset of rRNA genes without affecting the rRNA territories on other acrocentric chromosomes.

Our observation of the compartmentalized distribution of individual members of SNUL RNA within specific sub-nucleolar regions challenges the current view that all the nucleoli within a single nucleus are composed of identical domains. Future work will entail determining the mechanism(s) underlying

the constrained formation of SNUL territories and allele-specific spreading and regulation on autosomal regions.

## Limitations of the present study

Presently, very little is known about the sequences in the short arms of NOR-containing chromosomes. A recent study utilizing long-read sequencing in a haploid cell line revealed that p-arms of NOR-containing chromosomes are enriched with repeat sequences (*Nurk et al., 2022*). Higher levels of sequence similarity observed between *SNUL-1* candidates, and pre-rRNA made it impossible for us to precisely map the genomic coordinates of *SNUL-1* genes from the available long-read sequencing data set. It is intriguing to note that the SNUL-1 candidates showed the highest levels of sequence similarity with 21 S pre-rRNAs. Interestingly, many 'adenosine and Guanosine' sequence mismatches were observed between SNUL-1 candidates and 21 S pre-rRNA. We cannot rule out the possibility that SNUL-1 RNAs represent pre-rRNA variants that are extensively modified via post-transcriptional processes such as adenosine-to-inosine (A-to-I-editing). Earlier studies have reported the enrichment of A-to-I editing enzymes such as ADARs (Adenosine deaminase that acts on RNA) in the nucleolus and RNA hyper-editing is utilized as a mechanism of RNA nuclear-retention (*Sansam et al., 2003*; *Prasanth et al., 2005*; *Chen et al., 2008*; *Zhang and Carmichael, 2001*). SNUL clouds are formed due to the hyper-editing of pre-rRNA variants to perform regulatory functions in the nucleolus. Future studies will test this model. Complete genome assembly of p-arms from SNUL-1-expressing diploid cells would be essential to map *SNUL-1* genes in the genome and to identify the regulatory elements controlling the monoallelic expression of *SNULs*. Genomic annotation of the full-length *SNUL-1* genes are also crucial for designing strategies to specifically alter the expression of individual *SNUL* genes without targeting other SNUL-like genes, furthering mechanistic understanding of SNUL functions. Even with these technical limitations, the current study is highly impactful because our observations of the association of autosome arms by SNULs support a paradigm-shifting model that ncRNA-coating of chromosomes and their roles in gene repression is not restricted only to sex chromosomes. In addition, our study will serve as a starting point towards understanding how differential rDNA expression is achieved during physiological processes. Altogether, this study will form the basis for an entirely new avenue of investigation, which would help to understand the role of ncRNAs on monoallelic changes in autosomal chromatin structure and gene expression in the nucleolus.

# Materials and methods
## Cell culture

WI-38 and IMR-90 cells were grown in MEM medium supplemented with 10% fetal bovine serum (FBS), non-essential amino acid, sodium-pyruvate. HeLa and U2OS cells were grown in DMEM medium supplemented with 5% FBS. HTD114 cells were grown in DMEM medium supplemented with 10% FBS. hTERT-RPE1, mouse A9 +H15 and SH-SY5Y cells were grown in DMEM/F12 medium supplemented with 10% FBS. GM12878 cells were grown in RPMI1640 medium supplemented with 15% FBS. PC-3 cells were grown in RPMI-1640 medium supplemented with 10% FBS. MDA-MB-231 and LNCaP cells were grown in RPMI 1640 media supplemented with 10% FBS. SaOS-2 cells were grown in McCoy's 5 A medium supplemented with 15% FBS. MCF10A cells were grown in DMEM/F12 medium supplemented with 5% house serum, hydrocortisone, cholera toxin, insulin, and EGF. HS578 cells were grown in DMEM medium supplemented with 10% FBS and insulin. MCH065 cells were grown in DMEM medium supplemented with 10% FBS. All media were supplemented with Penicillin/Streptomycin. H9 hESCs and MCH2-10 iPSCs were grown on acid-treated coverslips coated with Matrigel hESC-Qualified Matrix (Corning, Product Number 354277) in mTeSR Plus (STEMCELL Technologies, Catalog #100–0276). Cells were maintained in a 5% $CO_2$ incubator at 37°C. Cell lines are obtained from commercial vendors such as ATCC and Coriell. Cell lines used in our study has been authenticated by STR profiling (UIUC Cancer Center). All cell lines were checked for mycoplasma.

## Transfection and virus infection

For ASO treatments, Ctr-ASO or ASO-SNUL were transfected to cells at a final concentration of 100 nM using Lipofectamine RNAiMax Reagent (Invitrogen). Cells were cultured for another 3 days before harvest.

pHAGE-mNG-C1-FBL and pHAGE-mTagBFP2-C1-B23 plasmids were gifts from Dr. Ling-ling Chen's lab *Yao et al., 2019*. HeLa cells in 3.5 cm dish were transfected with 500 ng of pHAGE-mNG-C1-FBL and/or pHAGE-mTagBFP2-C1-B23 using Lipofectamine 3000 Reagent (Invitrogen).

pCHA-hTAF1A-D (SL1 complex) plasmids were gifts from Dr. Kyosuke Nagata's lab (*Murano et al., 2014*). A9 +H15 cells in 3.5 cm dish were transfected with pCHA-hTAF1A-D plasmid, 400 ng per each plasmid, using Lipofectamine 3000 Reagent (Invitrogen).

For stably expressing mNG-FBL in WI-38 cells, lentiviral particles were packaged by transfecting pHAGE-mNG-C1-FBL, pMD2.G, psPAX2 to 293T cells in WI-38 growing medium. Virus was collected twice at 48 hr and 72 hr after transfection, removed of cell debris by centrifuge, and snap frozen. WI-38 cells were infected by the virus for 2 days and changed back to medium without virus.

## Transcription inhibition and epigenetic marker inhibition

For RNA Pol I inhibition, cells were treated with 1 µM BMH21 (Selleckchem), 10 ng/ml ActD (Sigma-Aldrich), or 1 µM CX5461 (Sigma-Aldrich) for 45 min or 2 hr. For RNA Pol II inhibition, cells were treated with (1) 5 µg/ml ActD for 2 h, (2) 2.5 µM Flavopiridol (Selleckchem) for 3 hr, or (3) 32 µg/ml DRB for 3 hr. After 3 hr of DRB treatment, cells were washed with PBS for 3 times and were recovered in fresh growth medium for 30 min or 60 min. For epigenetic mark inhibition, cells were treated with 80 nM TSA and 500 nM 5-Aza-dC for 6 days.

## RNA-fluorescence in situ hybridization (FISH)

For all the FISH and Immunofluorescence staining done with adherent cells, cells were seeded on #1.5 coverslips at least two days before experiments. For GM12878 and isolated HeLa nucleoli, suspension was smeared onto the Poly-L-lysine-coated (Sigma-Aldrich) coverslips prior to fixation.

For RNA-FISH using probes prepared by nick translation, cells were fixed by 4% PFA for 15 min at room temperature (rt) and permeabilized with 0.5% Triton X-100 for 5 min on ice. Alternatively, cells were pre-extracted by 0.5% Triton X-100 in CSK buffer for 5 min on ice and then fixed by 4% PFA for 10 min. Probes were made using Nick Translation Kit (Abbott Molecular) as per manufacturer's instructions, added to the hybridization buffer (50% formamide, 10% dextran sulfate in 2XSSC supplemented with yeast tRNA), and before hybridization. Hybridization was carried out in a humidified chamber in the dark overnight at 37°C. The coverslips were then washed in 2 X SSC and 1 X SSC and 4 X SSC. DNA is counterstained with DAPI. Coverslips were mounted in VectaShield Antifade Mounting Medium (Vector Laboratories) or ProLong Diamond Antifade Mountant (Invitrogen). Please see *Supplementary file 4* for primer and probe sequence details.

5'ETS smFISH probe sets were described in *Yao et al., 2019*. SNRPN and SPA2 smFISH probe sets were designed using Stellaris Probe Designer. Oligonucleotides with 3' amino group (LGC Biosearch Technologies) were pooled and coupled with either Cy3 Mono NHS Ester (GE healthcare) or Alexa Fluor (AF) 647 NHS Ester (Invitrogen) by incubation overnight at 37°C in 0.1 M NaHCO$_3$. Probes were then purified by G-50 column (GE Healthcare) and ethanol precipitation. Concentration was measured by the OD at 550 nm (Cy3) or 650 nm (AF647). For RNA-FISH involving smFISH probes, smFISH probes were added to Stellaris RNA FISH Hybridization Buffer (LGC Biosearch Technologies) with 10% formamide at a final concentration of 125 nM. Hybridization was carried out in a humidified chamber in the dark for 6 hr at 37°C . The coverslips were then washed with Stellaris RNA FISH Wash Buffer A and mounted as described above.

Digoxin-labeled RNA probes were in vitro transcribed as per manufacturers' instructions (DIG RNA labeling Mix, Roche; T7 polymerase, Promega; SP6 Polymerase, Promega) and purified by G-50 column (GE Healthcare). For RNA-FISH using ribo-probes, cells on coverslips were fixed by 4% PFA for 10 min at rt, and then treated with 0.25% acetic anhydride in 0.1 M triethanolamide (pH 8.0) for 10 min. Coverslips were washed in 1XSSC for 5 min, treated with 0.2 N HCl for 10 min, and pre-hybridized in 50% formamide, 5XSSC for at least 6 hr at rt. Dig-labeled RNA probes were added to the hybridization buffer (50% formamide, 5XSSC, 1 X Denhardt's solution, 0.1% Tween20, 0.1% [w/v] CHAPS, 100 µg/ml Heparin, 5 mM EDTA, and 50 µg/ml Yeast tRNA) at a final concentration of 2 µg/ml. Hybridization was carried out in a humidified chamber in the dark overnight at 50°C. The coverslips were then washed with 0.2XSSC for 1 hr at 55°C, blocked in 4% BSA, PBS for 30 min at 37°C, and incubated with anti-Dig-FITC or -Rhodamine (1:200) (Roche) in 1% BSA, PBS for 1 hr at 37°C. The

coverslips were washed twice with washing buffer (0.1% Tween20, 2XSSC) and refixed with 4% PFA for 15 min at rt.

For RNase A treatment, pre-extracted cells were incubated with 1 mg/ml RNase A in CSK buffer for 30 min at 37°C. Cells were then fixed by 4% PFA for 15 min at rt and processed to RNA-FISH. For DNase I treatment, fixed and permeabilized cells were incubated with 200 U/ml DNase I (Sigma) in DNase I buffer prepared with PBS for 2 hr at 37°C, followed by incubation in Stop solution for 10 min at room temperature. RNA-FISH was then performed as described above.

## RNA-DNA FISH

For DNA-FISH using chromosome paint probes (Chrs. 13, 15, 22) (MetaSystems), after fixation and permeabilization, coverslips were incubated in 20% glycerol overnight and then went through freeze-thaw by liquid nitrogen for at least six cycles. Coverslips were then treated with 0.1 N HCl for 5 min and prehybridized in 50% formamide, 2XSSC for 30 min at rt. Probe mix was made by adding the RNA-FISH probe into the chromosome paint probe. Probes were applied to the coverslips and denatured with the coverslips at 75–80°C on a heating block. Hybridization was carried out in a humidified chamber in the dark for 48 hr at 37°C.

For FISH using DNA-FISH probes made by nick translation, cells were pre-extracted and fixed. Salmon sperm DNA and Human Cot-I DNA were added to the hybridization buffer. Denaturation and hybridization were performed as described above.

## DNA-FISH on metaphase spread

Cells were grown to ~70% confluence and treated with KaryoMax Colcemid solution (Gibco) at a final concentration of 0.1 µg/ml in growth medium for 3 hr. Mitotic cells were then shaken off and pelleted by centrifuge. Cells were then gently resuspended in 75 mM KCl and incubated at 37°C for 30–40 min. Cells were then fixed by freshly prepared fixative (methanol: acetic acid 3:1 [v/v]) and dropped onto pre-cleaned microscope slides from height. After air-drying, slides were stored at –20°C for a least overnight before the DNA-FISH. For the DNA-FISH on metaphase chromosomes, slides were rehydrated with PBS and then treated with 50 µg/ml Pepsin in 0.01 N HCl at 37°C for 9 min. Slide were then rinsed by PBS and 0.85% NaCl sequentially and dehydrated by a series of Ethanol at different concentration (70%, 90%, and 100%). Air-dried slides were then subjected to hybridization as described above.

## Immunofluorescence staining (IF)

Cells on coverslips were fixed and permeabilized before blocking in 1% BSA for 30 min at rt. Coverslips were then incubated with primary antibodies (anti-FBL, 1:500, Novus Biologicals, NB300-269; anti-RPA194, 1:50, Santa Cruz, sc-48385; anti-UBF, 1:50, Santa Cruz, sc-13125; anti-DDX21, 1:20000, proteintech,10528–1-AP). and secondary antibody (anti-mouse IgG2a AF647, 1:2000, Invitrogen, A-21241; anti-mouse IgG AF568, 1:2000, Invitrogen, A-11031; anti-mouse Cy5, 1:1000), sequentially. Coverslips were then washed with PBS and refixed with 4% PFA. RNA-FISH was then carried out if needed.

## 5-FU metabolic labeling

Cells were grown to ~70% confluence on the day of experiments. Cells were treated with 2 mM 5-FU (Sigma-Aldrich, F5130) for specified time periods before harvest. To detect incorporated 5-FU, IF was performed with anti-BrdU antibody (1:800, Sigma-Aldrich, B9434) as described above.

## Nucleoli isolation

HeLa nucleoli were isolated as described in *Bai and Laiho, 2015* with adjustments. Briefly, HeLa cells were collected by trypsinization and lysed in nuclear extraction buffer (50 mM Tris-HCL, pH7.4, 0.14 M NaCl, 1.5 mM MgCl$_2$, 0.5% NP-40, 1 mM DTT, and RNase Inhibitor). Nuclei was precipitated and resuspended in S1 solution (0.25 M sucrose and 10 mM MgCl$_2$). Nuclear suspension was gently layered on S2 solution (0.35 M sucrose and 0.5 mM MgCl$_2$) and spun at 2000 × *g* for 5 min at 4°C. Purified nuclei were then sonicated by Bioruptor UCD-200 at high mode until nucleoli were released. Another sucrose cushion was then carried out with S3 solution (0.88 M sucrose and 0.5 mM MgCl$_2$).

Isolated nucleoli were then resuspended in S2 and subjected to RNA extraction by Trizol Reagent (Invitrogen) or RNA-FISH.

## Northern blot

For the pre-rRNA Northern, 2 µg of total RNA extracted from WI-38 cells treated with Ctr-ASO or ASO-SNUL was separated on 1% denature agarose gel prepared with NorthernMax Denaturing Gel Buffer (Ambion) and run in NorthernMax Running Buffer (Ambion). RNA was then transferred to Amersham Hybond-N+blot (GE Healthcare) by capillary transfer in 10 x SSC and crosslinked to the blot by UV (254 nm, 120mJ/cm$^2$). The DNA probes were labeled with [α–32P] dCTP by Prime-It II Random Primer Labeling Kit (Stratagene) as per manufacturer's instructions. Hybridization was carried out using ULTRAhyb Hybridization Buffer (Ambion) containing 1X10$^6$ cpm/ml of denatured radiolabeled probes overnight at 42°C . Blots were then washed and developed using phosphor-imager.

## Native RNA immunoprecipitation

WI38 cells were washed twice with cold PBS and collected by centrifuge (1000 × g, 10 mins at 4°C ). Cells were then lysed in 2 ml RIP buffer (50 mM Tris pH 7.4, 150 mM NaCl, 0.05% Igepal, 1 mM phenylmethyl sulfonyl fluoride [PMSF], 1 µM Leupeptin, 1 µM Pepstatin, 0.2 µM Aprotinin, and 2 mM VRC[NEB]), and sonicated by Bioruptor UCD-200 at high mode on ice. Cells were centrifuged at 1000 × $g$ at 4°C for 10 min and supernatants were then pre-cleared with15 µl Dynabeads Protein G (Invitrogen) for 30 min. FBL antibody (Abcam) or rabbit IgG2b was incubated with 25 µl Dynabeads Protein G for 30 min. The cell supernatants were then incubated with Dynabeads Protein G at 4°C for 2 hr. The Dynabeads Protein G were then washed with high salt buffer (50 mM Tris pH 7.4, 650 mM NaCl, 0.15% Igepal, 0.5% sodium deoxycholate,1 mM PMSF, 1 µM Leupeptin, 1 µM Pepstatin, 0.2 µM Aprotinin, and 2 mM VRC[NEB]) for three times and RIP buffer twice, followed by RNA isolation and RT-qPCR.

## Histone mark staining in mitotic spreads

HTD114 and RPE1 cells were treated with colcemid for 1 hr and collected by centrifuge (1500 rpm, 10 min at RT). Cells were then suspended in 0.06 M KCl buffer at 37°C for 15 min. The cytospin samples were made by centrifuging for 10 min at 1000 rpm with Cytospin 4 (Thermo Fisher) on Superfrost Plus slides. The slides were fixed with 2% PFA for 15 min, and followed by blocking with 3% BSA and 0.1% Tween in PBS at 37°C for 15 min. The slides were then incubated with histone mark primary antibodies (anti-H3Ac, 1:100, Sigma-Aldrich, 06–599; anti-H3K4me3, 1:100, Sigma-Aldrich, 07–473; anti-H4Ac, 1:100, Sigma-Aldrich, 06–598) and secondary antibody (anti-rabbit IgG AF 488 or anti-rabbit IgG AF 658, 1:100) sequentially. Slides were washed with PBS, refixed with 2% PFA for 10 min, and then fixed with 3:1 methanol:acetic acid at –20°C for 15 min followed by air dry. Air-dried slides were then subjected to DNA-FISH as described above.

## Thymidine bock and EdU staining

HTD114 cells were seeded on coverslips and blocked with 2 mM Thymidine for 24 hr, released for 12 hr and then blocked again with 2 mM Thymidine for 24 hr and release at different time points. 125 µMEdU was added to the cells 30 min before harvest. The cells were fixed with 4% PFA for 10 min and permeabilized with 0.5% Triton in PBS on ice for 10 min. Click-reaction was performed by incubating the coverslips with 2 mM CuSO$_4$, 100 mM sodium ascorbate, 2 µM Alexa Fluor 488 Azide (Invitrogen) in PBS for 30 min. The coverslips were then subjected to DNA-FISH as described above.

## Replication timing-specific hybridization (ReTiSH)

HTD114 and WI38 cells were labeled with 30 µM BrdU for 5,6 or 14 hr. A total of 0.1 µg/ml Colcemid was added to the cells 1 hr before harvest. Cells were harvested and incubated in 0.075 M KCl for 40 min and fixed with 3:1 methanol:acetic acid on ice for three times, 10 min each. The fixed cells were dropped on ethanol treated wet slides and air dried. The air-dried slides were treated with 100 µg/ml RNase A at 37°C for 10 min and refixed with 2%PFA for 5 min, incubated with pepsin (0.5 mg/ml in 1 N HCl) for 10 min at 37°C and then treated with 0.5 µg/µl Hoechst 33258 for 15 min. The slides were then flooded with 2XSSC and exposed to 350 nm UV light for 30 min, followed by incubating with 3 U/µl Exo III for 15 min at 37°C. Slides were then dehydrated by a series of Ethanol at different concentration

(70%, 85%, and 100%) and air dry. Air-dried slides were then hybridized with denatured DNA probes for 24 hr at 37°C and washed with 2xSSC for three times at 42°C followed by DAPI staining.

## RNA FISH flow

Five million RPE1 cells were trypsinized and collected by centrifuging at 300 × *g*, 5 min at RT. The cell pellet was resuspended twice in PBS, pre-extracted by 0.5% Triton X-100 in CSK buffer for 10 min on ice, pelleted and then fixed by 4% PFA for 10 min at 37 . The fixed cells are pelleted and washed with PBS and incubated in 70% ethanol at −20°C overnight. The cells were resuspended in hybridization buffer with nick translated probes and incubated overnight at 37°C. After hybridization, the cells are centrifuged and washed with 2XSSC for three times at 42°C, 10 min each. The cells are then resuspended in 2XSSC with DAPI. The fluorescence intensity is measured using BD LSRII flow cytometer and analyzed using the FCS Express 6.0.

## Imaging acquisition

For widefield microscopy, z-stack images were taken using either (1) DeltaVision microscope (GE Healthcare) equipped with 60 X/1.42 NA oil immersion objective (Olympus) and CoolSNAP-HQ2 camera, or (2) Axioimager.Z1 microscope (Zeiss) equipped with 63 X/1.4 NA oil immersion objective and Zeiss Axiocam 506 mono camera. Images were then processed through deconvolution and maximum intensity projection.

SIM images were taken using either DeltaVision OMX V3 system (GE Healthcare) equipped with a 100 X/1.4 NA oil immersion objective, three laser beams (405 nm, 488 nm, and 568 nm) and EMCCD camera (Cascade II 512), or SR-SIM Elyra system (Zeiss) equipped with 63 X/1.4 NA oil immersion objective, four laser beams (405 nm, 488 nm, 561 nm, and 642 nm). For DeltaVision OMX V3 system, Channels were aligned for each of the experiments using the registration slide and TetraSpeck microspheres (Invitrogen). SIM image stacks were acquired with a z-interval of 0.125 μm, five phases and three angles. SIM reconstruction and registration of channels were performed by softWoRX software (GE Healthcare). For SR-SIM Elyra system, channels were aligned for each of the experiments using the TetraSpeck microspheres (Invitrogen). SIM image stacks were acquired with a z-interval of 0.125 μm, five phases and three rotations. SIM reconstruction and channel alignment were performed by ZEN 2011 software (Zeiss).

## Imaging analyses

For colocalization analyses, 3D SIM stacks were imported into Fiji/ImageJ. The nucleolar area containing SNUL-1 signal was selected and Pearson's correlation coefficients (no threshold) were calculated by the Coloc2 Plugin.

For the measurement of integrated density, z-stacks were imported into Fiji/ImageJ and maximum intensity projection was performed. Signal of interest was then segmented by Maximum Entropy Multi- Threshold function in the ij-Plugins Toolkit with number of thresholds = 3. A Binary mask was generated based on the second level of the threshold from the last step. The integrated density of the signal of interest from the original image within the mask was then measured. For rDNA contents on the two Chr. 15 alleles in WI38 cells, relative integrated density was calculated by dividing the measurement of the rDNA signal on the Chr. 15 with the larger 15Sat III by the measurement on the other Chr. 15. For rDNA contents on the two Chr. 15 alleles in hTERT-RPE1 cells, relative integrated density was calculated by dividing the measurement of the larger rDNA signal by the measurement of the smaller rDNA signal. For 15Sat III and 15CEN, relative integrated density was calculated by dividing the measurement of the larger signal spot by that of the other spot signal within the same cell. For ReTiSH in HTD114 and WI38 control and SNUL-depleted cells, relative integrated density between two alleles was calculated by dividing the measurement of the rDNA signal on the Chr. 15 with the larger CEN15 by the measurement on the other Chr. 15, and by dividing the measurement of the rDNA signal on the Chr. 15 with the larger 15Sat III by the measurement on the other Chr. 15, respectively.

For the measurement of 5-FU incorporation and 5'ETS signal in control and SNUL-depleted cells, z-stacks were imported into Fiji/ImageJ and maximum intensity projection was performed. Nuclei were segmented by optimized threshold and inverted into binary mask. The integrated density of 5-FU signal or 5'ETSsignal in each of the nuclei was measured.

For the measurement of 5'ETS-1 signal intensity in FC of nucleolus in control and SNUL-depleted cells, the middle z session was selected for each image and imported into Fiji/ImageJ then split into single channels. FCs were segmented by auto threshold of RPA194 channel and inverted into binary mask. The binary masks were applied to the 5'ETS-1 channel and the integrated intensity of 5'ETS-1 signals within FCs were counted for each nucleolus. The relative 5'ETS-1 signal intensity in FCs was calculated by dividing the integrated intensity of 5'ETS-1 signals within FCs by the integrated intensity of the entire image.

For the measurement of FBL and RPA194 signal distribution, 20 of DFC and FC units were selected and the middle z session for each unit was chosen. Averaged images were generated by applying Average Intensity Projection in Fiji/ImageJ. FBL or RPA194 signal was segmented by optimized threshold and inverted into binary mask. The binary masks were applied to FBL or RPA194 channel and the area was counted for each FC/DFC unit.

For the measurement of histone mark intensities in HTD114 and RPE1 control and SNUL-depleted cells, z-stacks were imported into Fiji/ImageJ and maximum intensity projection was performed. rDNA signal on Chr.15 alleles was segmented by optimized threshold and inverted into binary mask. The binary masks were applied to the histone mark channel and the integrated intensity of histone mark signals was counted for each metaphase spread. The relative integrated density between two alleles was calculated by dividing the measurement of the rDNA signal on the Chr. 15 with the larger CEN15 by the measurement on the other Chr. 15, and by dividing the measurement of the rDNA signal on the Chr. 15 with the larger 15Sat III by the measurement on the other Chr. 15, respectively.

## PacBio iso-seq

Total RNA from isolated HeLa nucleoli was poly-adenylated by Poly(A) Polymerase Tailing Kit (Epicentre) and depleted of rRNA by the RiboMinus Human/Mouse Transcriptome Isolation Kit (Invitrogen). RNA was then reverse transcribed by the SMARTer PCR cDNA Synthesis Kit (Clontech) and amplified for 15 cycles using KAPA HiFi PCR Kit (KAPA biosystems). cDNA was then separated into two fractions by size using 0.5 X and 1 X AMPure PB Beads (Pacific Scientific), respectively. SNUL-1 was then enriched from the two fractions by xGen capture procedure with SNUL-1 Probe 4 using the xGen hybridization and Wash Kit (IDT). Another round of PCR amplification was carried out after the capture. The two fractions were then combined. Library was prepared by Amplicon SMRTbell Prep (Pacific Scientific) and sequenced on LR SMRT cell with 20 hr movie.

## Nanopore sequencing

Total RNA from isolated HeLa nucleoli was depleted of rRNA by the RiboMinus Human/Mouse Transcriptome Isolation Kit (Invitrogen). RNA was converted to double stranded cDNA using random hexamer with the NEBNext Ultra RNA First Strand and NEBNext Ultra RNA 2nd Strand Synthesis Kits (NEB). 1D library was prepared with the SQK-LSK108 kit (Oxford Nanopore) and sequenced on a SpotON Flowcell MK I (R9.4) flowcell for 14 hr using a MinION MK 1B sequencer. The flowcell was washed and another identical library was loaded in the same flowcell and sequenced for another 14 hr. Basecalling was performed with Albacore 2.0.2.

The PacBio Iso-seq and nanopore RNA-seq data sets are deposited to the NCBI SR data base. The Bioproject accession number is SRA data: PRJNA814414.

## Sequencing analyses

To find transcripts that are similar to SNUL in the high-quality PacBio database, RIBlast (*Fukunaga et al., 2017*), a tool for predicting RNA-RNA interactions, is used in its default setting for SNUL-1 probe 1 as the query RNA. The top 2000 candidates with the lowest interaction energy were intersected to generate the final set of 507 transcripts like SNUL-1. Pairwise BLASTs of each of the top ranked PacBio Iso-Seq clones and genome assembly hg38 were performed to pick the candidates showing the least similarity with any of the annotated genes. A reference fasta file was generated with the top 5 picked candidates with least similarity (BLAST) to the annotated genes from PacBio sequencing dataset. The fast5 file from nanopore long read sequencing was basecalled using Guppy V4.2.2.A reference fasta file was generated from the top 5 sequences identified previously from BLAST-based least similarity to annotated genes. The fastq file from Nanopore sequencing containing 1,267,135 reads (across two runs) was aligned to the reference fasta file, generated from top 5 candidates from PacBio

sequencing data, using minimap2 (*Li and Birol, 2018*). Alignment statistics were computed using samtools – flagstat option (*Li et al., 2009*), a total of 123,582 (9.7%) were mapped to the reference fasta file. Mapped reads were extracted into a sam file and indexed using samtools, for visualizing the alignments through IGV (Integrative Genomics Viewer) (*Zentner et al., 2011*). To generate an accurate version of SNUL-1 transcripts, we generated a consensus sequence from the above Nanopore based long-read alignments to the PacBio based candidate sequences, using samtools and bcftools (*Danecek and McCarthy, 2017*). All the reads aligning to the SNUL-1 transcripts were extracted and a consensus sequence from these reads was assembled using the combination of modules mpileup, call, and vcfutils from samtools and bcftools as depicted in Figure S1g. To evaluate the specificity of the assembled transcripts, we performed a similarity comparison between the generated consensus sequence against rRNA and PacBio Iso-Seq CS clones. We observed that the sequences identified/ generated from our analysis were more analogous to the Iso-Seq clones over rRNA.

To verify the error rate of PacBio sequencing technology, for each isoform in the high-quality PacBio database, we ran BLAST against the human transcript database (GRCh38.p13 assembly). The maximum number of target sequences and the maximum number of high-scoring segment pairs were set to 20 and 1 respectively, and the rest of the arguments were set to default in the BLAST runs. LAGAN-v2.0 (*Brudno et al., 2003*) was then used to perform pairwise global alignment between each isoform and its corresponding top 20 best matches, found by BLAST. A dissimilarity score was assigned to each matched candidate by taking the ratio of mismatching sites to all the sites where both isoform and the matched candidate did not contain gaps. The matched candidate with the least dissimilarity score was taken as the best match to the isoform. The mean of the dissimilarity scores, associated with isoforms having GC content greater than 60% (matching the GC content of ITS1), being ~0.5% verifies the PacBio error rate (<1%).

To determine whether the five isoforms capable of detecting SNUL-1 are different transcripts, and their difference is not due to sequencing error, we propose the following three hypotheses to be tested:

$H_0$: There is one known gene whose transcripts are $I_1,…,I_5$.

$H_1$: There is one unannotated gene, that is with no transcripts present in human transcript database, whose transcripts are $I_1,…,I_5$.

$H_2$: There are multiple unannotated/annotated genes whose transcripts are $I_1,…,I_5$.

If $H_0$ is true, then there exists a known transcript such that the dissimilarity score ($DS_i$) between isoform $i$ ($I_i$) and that transcript follows the empirical distribution of the dissimilarity scores in Figure S1l. Thus, we calculated an empirical p-value for $DS_i$ (*Supplementary file 2*) and performed Fisher's combined probability test on the five p-values corresponding to the five isoforms, obtaining a p-value <0.000002 ($\chi^2$ = 46, df = 10), suggesting that $H_0$ does not hold.

If $H_1$ is true, there should be one unannotated transcript whose dissimilarity score with each read is about 0.5% (the empirical mean of the dissimilarity scores). Therefore, the pairwise dissimilarity scores for the isoforms should be about 1%. We computed the real pairwise dissimilarity by doing global pairwise alignment for each pair of isoforms using LAGAN (*Supplementary file 4*). Intuitively, the observed pairwise dissimilarities being 4% or above suggest that $H_1$ is not true. We also provide a rigorous statistical analysis to reject $H_1$. First, we approximate the empirical distribution of the dissimilarity scores with an exponential probability density with mean 0.5%, that is 0.005. Second, we assume that the dissimilarity scores of the isoforms are independent and identically distributed (i.i.d.) random variables that is $\forall i \in \{1,…,5\}, DS_i \sim exp(\lambda), \lambda = 200$ (since mean of exponential distribution is $1/\lambda$). Given the i.i.d. assumption, the maximum pairwise dissimilarity for each pair of isoforms $I_i$ and $I_j$, $DS_{ij}^{max} = DS_i + DS_j$, has Erlang distribution with shape parameter 2 and rate parameter 200, as the sum of two independent exponential random variables with the same rate parameter has an Erlang distribution with scale parameter equal to that common rate. This allows us to estimate a p-value for each of the $\binom{5}{2} = 10$ observed pairwise dissimilarity scores $DS_{ij}$ (*Supplementary file 4*). Subjecting the 10-resulting p-values to a Fisher's combined probability test, we obtained a p-value <1E-21 ($\chi^2$ =154 df=20). This allowed us to reject the $H_1$ hypothesis, leading to accepting the $H_2$ hypothesis.

For the analyses of alignment between SNUL-1 candidates and pre-rRNA, the candidate sequences were aligned to the canonical 21 S sequence and to each other using the LAST algorithm as described previously (*Kiełbasa et al., 2011*).

## Data analyses and statistics

The data used in this study are performed at least biological triplicates. Statistical analyses (two-tailed Student's t-test and Mann-Whitney test) were done by GraphPad Prism. Please see figure legends for details about the sample sizes and statistical significance.

## Acknowledgements

We thank members of Prasanth's laboratory and Dr. Ashish Lal (NCI, NIH) for their valuable comments. We thank Drs. Eric Bolton (UIUC) for sharing prostate cancer cell lines (PC3 and LNCap); Drs. Mathew Thayer and Leslie Smith (OHSU) for HTD114 cells and for help with the ReTiSH; Dr. Andrew Belmont (UIUC) for human stem cells; Dr. Alok Sharma and Dr. Drinda Swanson from Abbott Inc.Inc for CEN15 and 15Sat III DNA probes; Prof. Ling-ling Chen (SIBCB) for lentiviral constructs expressing fluorescently-labeled nucleolar proteins; Prof. Kyosuke Nagata (University of Tsukuba) for SL1 plasmids and Prof. Vikram Paralkar and Charles Antony (UPenn) for help with FISH-FLOW. We thank Dr. Jason Underwood (PacBiosciences) for the help to perform Iso-Seq. This work was supported by National Institute of Health R21-AG065748 & R01-GM132458 to KVP, GM125196 to SGP, R35GM131819 to SS, R21AG065748 and R01GM123314 to SCJ, Cancer center at Illinois seed grants and Prairie Dragon Paddlers to KVP, National Science Foundation (NSF) to KVP [1723008 {EAGER}] & 2243257 {NSF center for quantitative cell biology}, SGP [1243372, 1818286, 2225264] and SCJ [1940422 &1908992]. HJ acknowledges support from the NIH (R01-GM120552). SMF is an employee of Ionis Inc.Inc, and ET and JK work for PACBIO and receive salary from the respective companies.

## Additional information

### Competing interests

Elizabeth Tseng, Jonas Korlach: affiliated with Pacific Biosciences. The author has no financial interests to declare. Susan M Frier: affiliated with Ionis Pharmaceuticals Inc. The author has no financial interests to declare. The other authors declare that no competing interests exist.

### Funding

| Funder | Grant reference number | Author |
| --- | --- | --- |
| National Institute of General Medical Sciences | GM132458 | Kannanganattu V Prasanth |
| National Institute on Aging | AG065748 | Kannanganattu V Prasanth |
| National Institute of General Medical Sciences | GM125196 | Supriya G Prasanth |
| National Institute of General Medical Sciences | R35GM131819 | Saurabh Sinha |
| National Institute on Aging | R21AG065748 | Sarath Chandra Janga |
| National Institute of General Medical Sciences | R01GM123314 | Sarath Chandra Janga |
| National Science Foundation | 1723008 | Kannanganattu V Prasanth |
| National Science Foundation | 1243372 | Supriya G Prasanth |
| National Science Foundation | 1818286 | Supriya G Prasanth |
| National Science Foundation | 2225264 | Supriya G Prasanth |
| National Science Foundation | 1940422 | Sarath Chandra Janga |

| Funder | Grant reference number | Author |
| --- | --- | --- |
| National Science Foundation | 1908992 | Sarath Chandra Janga |
| National Institute of General Medical Sciences | GM120552 | Hong Jin |
| National Science Foundation | 2243257 | Kannanganattu V Prasanth |

The funders had no role in study design, data collection and interpretation, or the decision to submit the work for publication.

## Author contributions

Qinyu Hao, Resources, Data curation, Formal analysis, Validation, Visualization, Methodology, Writing – original draft; Minxue Liu, Conceptualization, Data curation, Formal analysis, Validation, Investigation, Visualization, Methodology; Swapna Vidhur Daulatabad, Resources, Data curation, Software, Formal analysis, Investigation, Methodology; Saba Gaffari, Resources, Data curation, Software, Formal analysis, Methodology; You Jin Song, Validation, Investigation, Methodology; Rajneesh Srivastava, Resources, Data curation, Software, Formal analysis; Shivang Bhaskar, Xin Chen, Chengliang Wang, Methodology; Anurupa Moitra, Formal analysis, Methodology; Hazel Mangan, Rachel B Gilmore, Hong Jin, Jonas Korlach, Resources, Methodology; Elizabeth Tseng, Data curation, Formal analysis, Methodology; Susan M Frier, Sui Huang, Stormy Chamberlain, Resources; Brian McStay, Resources, Formal analysis, Writing – original draft; Saurabh Sinha, Software, Supervision, Methodology; Sarath Chandra Janga, Resources, Software, Methodology; Supriya G Prasanth, Supervision, Investigation, Writing – original draft; Kannanganattu V Prasanth, Conceptualization, Supervision, Funding acquisition, Investigation, Writing – original draft, Project administration, Writing – review and editing

## Author ORCIDs

Qinyu Hao ⓘ http://orcid.org/0000-0002-7059-7741
Minxue Liu ⓘ http://orcid.org/0000-0002-7705-5822
Swapna Vidhur Daulatabad ⓘ http://orcid.org/0000-0001-5288-8599
You Jin Song ⓘ http://orcid.org/0000-0002-2060-3429
Rajneesh Srivastava ⓘ http://orcid.org/0000-0003-1093-2082
Sarath Chandra Janga ⓘ http://orcid.org/0000-0001-7351-6268
Supriya G Prasanth ⓘ https://orcid.org/0000-0002-3735-7498
Kannanganattu V Prasanth ⓘ https://orcid.org/0000-0003-4587-8362

## Decision letter and Author response

Decision letter https://doi.org/10.7554/eLife.80684.sa1
Author response https://doi.org/10.7554/eLife.80684.sa2

## Additional files

### Supplementary files

• Supplementary file 1. Sequence of the SNUL-1 probe. Please see the Excel file

• Supplementary file 2. The CS candidate sequences. Please see the Excel file

• Supplementary file 3. The dissimilarity score between each isoform and its best match from human transcript database. For each isoform the empirical p-value is the empirical probability of observing a dissimilarity score greater than or equal to its dissimilarity score. The product of the 5 p-values being in the order of $10^{-10}$ rejects the hypothesis of all isoforms being the transcripts of the same known gene.

• Supplementary file 4. The pairwise dissimilarity for the isoforms. The pairwise dissimilarity for the isoforms. Global pairwise alignment was used to compute the dissimilarity score for each pair of isoforms. The dissimilarity score was computed by taking the ratio of mismatching to matching sites where both isoforms do not contain gaps. For each pair of isoforms, the p-value shows the probability of observing a dissimilarity score greater than or equal to their dissimilarity by approximating the empirical distribution of pairwise dissimilarity with erlang distribution.

• Supplementary file 5. Probes used in this study. Please see the Excel file.

• MDAR checklist

## Data availability

Sequencing data have been deposited in the NCBI BioProject database under accession code PRJNA814414.

The following dataset was generated:

| Author(s) | Year | Dataset title | Dataset URL | Database and Identifier |
|-----------|------|---------------|-------------|-------------------------|
| Hao et al | 2023 | Monoallelically-expressed Noncoding RNAs form nucleolar territories on NOR-containing chromosomes and regulate rRNA expression | https://www.ncbi.nlm.nih.gov/bioproject/PRJNA814414 | NCBI BioProject, PRJNA814414 |

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
