## [Editor Report]

A long-standing question has been to understand which of the many ribosomal RNA clusters are expressed and how they are regulated. Here the authors characterize a new, noncoding RNA SNUL-1 and propose that its expression pattern in cis regulates ribosomal RNA expression.

---

## [Decision Letter]

**Decision letter after peer review:**

[Editors’ note: the authors submitted for reconsideration following the decision after peer review. What follows is the decision letter after the first round of review.]

Thank you for submitting the paper "Monoallelically-expressed Noncoding RNAs form nucleolar territories on NOR-containing chromosomes and regulate rRNA expression" for consideration by *eLife*. We apologize for the terrible delay in providing this decision letter. We had much difficulty securing reviews. In the end your article could only be reviewed by 1 peer reviewer, but the evaluation has been overseen by a Reviewing Editor and a Senior Editor. Another reviewer did not respond despite being chased for several weeks, and a second reviewer could not review in the end but provided informal remarks that we are passing onto you in case they would be helpful. The reviewers have opted to remain anonymous.

We are sorry to say that, after consultation with the reviewers, we have decided that this work cannot be published in *eLife* as it is.

The reviewer(s) found the paper interesting, but the main concerns are that the study does not provide enough information about the RNA and equally important, the manuscript did not discuss preceding work that impacts the interpretation and novelty of the findings. The observation of monoallelic expression from rRNA encoding loci is interesting, but it has been established in 2009 by the work, Allelic inactivation of rDNA loci. Genes Dev. 2009 Oct 15;23(20):2437-47. doi: 10.1101/gad.544509. You and your coauthors may not be aware of the paper, but there may be relevant data from this paper to help understand the NOR phenomenon better. We do acknowledge that you have succeeded in identifying and characterizing a new family of ncRNAs which seem to play a role in the mono-allelic expression of r-RNA. These ncRNAs are expressed in a mono-allelic manner. This remains interesting but we feel that the study would have to go further since monoallelic expression was established more than 10 years ago. Reviewer 1 also mentioned that the experiments would have to be better described for them to evaluate the findings.

We hope these comments will be helpful to you as you move forward. We hope that you will conduct further experiments that would address the above concerns.

Recommendations for the authors:

The manuscript "Monoallelically-expressed Noncoding RNAs form nucleolar territories on NOR-containing chromosomes and regulate rRNA expression" reports the discovery of a family of ncRNAs they call SNULs for Single NUcleolus Localized RNA and examine their localization with respect to nucleoli and reports that the RNAs they are examining are monoallelically expressed in a mitotically stable manner similar to what happens in X inactivation.

These RNAs come from a screen which is not well described and the descriptions of the sequence analyses are unclear, so it is difficult to know exactly what they are analyzing in the manuscript. If these are RNAs with reasonable abundance, then they should be findable without the extensive PCR amplification they appear to have done for the PacBio sequencing (the methods section is not clear on exactly how many rounds of PCR were performed). Moreover, given the acknowledged sequence similarities of the SNULs with other RNAs, the possibility of chimaera formation during PCR amplification is high. They are clearly detecting RNAs associated with nucleoli but exactly what they are examining is unclear. It is possible that a clear determination of the genomic origin of these RNAs will be complicated by the repetitive sequences in the regions of the genome where they reside.

Note also that the idea of monoallelic expression from rRNA encoding loci is interesting, but has been established in 2009. Title: Allelic inactivation of rDNA loci. Genes Dev. 2009 Oct 15;23(20):2437-47. doi: 10.1101/gad.544509.

If there are analyses which could be added to further clarify how the different types of sequencing complemented and/or reinforced one another, that might make the characterization of the RNAs being studied here more convincing.

[Editors’ note: further revisions were suggested prior to acceptance, as described below.]

Thank you for resubmitting your work entitled "Monoallelically-expressed Noncoding RNAs form nucleolar territories on NOR-containing chromosomes and regulate rRNA expression" for further consideration by *eLife*. Your revised article has been evaluated by Kevin Struhl (Senior Editor) and a Reviewing Editor, and we apologize for the long delay in the review process. As you know, we had difficulty obtaining reviewers in the first round and in the end, only had one reviewer who was able to turn in comments. Fortunately, we secured two additional reviewers in this round, although there was a delay in obtaining the last review. As you will see, all three reviewers view your paper positively and we would like to invite you to submit a revised manuscript based on their recommendations.

Specifically, we hope you can address these comments in a revised paper.

1. Reviewer 2 suggested visualizing the SNUL transcripts using a northern blot to give readers a sense of the length and abundance of the various isoforms. If you can provide this information from the Nanopore or PacBio data, please do so in lieu of the northern. (also see note from Reviewer 1 about presentation of the Nanopore sequencing data)

2. Reviewer 3 wished to see the location(s) of SNUL-1 and to which extent its effect on pre-rRNA processing can be distinguished from targeting the pre-rRNA itself. Please see what info you could provide in a revision.

3. Disambiguation of the SNUL-1 target from co-targeting the pre-rRNA itself.

4. Please also address the more specific and minor comments of the reviewers.

*Reviewer #1 (Recommendations for the authors):*

The manuscript by Hao et al., "Monoallelically-expressed Noncoding RNAs form nucleolar territories on NOR-containing chromosomes and regulate rRNA expression" characterizes novel non-coding RNA emanating from the rDNA regions of the human genome. The revised manuscript has thoughtfully considered the concerns raised in the initial review.

Comments / points to be addressed:

1. line 66/67, instead of "and were transcriptionally inactive" perhaps say something along the lines of "and had chromatin marks consistent with being transcriptionally inactive".

2. line 178/179. Please give a little more detail as to what you mean by "partially align to 18S" and "align to ITS1". What % alignment over what distance, or whatever makes sense given the nature of the alignments being mentioned?

3. line 242. Use of the word "biallelically" is confusing in this context because it seems that the text is referring to a second location which could be one of the two alleles of several rDNA loci.

4. lines 251 -258. The text is talking about the SNUL-1 cloud association with one allele of Chr. 15 NOR and then mentions the association rate with Chr. 13 and 22, but not the other allele of Chr. 15.

5. lines 285 – 287. What about mentioning if it could be observed whether the other Chr. 15 had transcriptionally active rDNA clusters?

6. lines 300 – 303. I suggest not using "non-randomly associates" as this is vague, and instead use terminology similar to what is used in the next paragraph where the ideas of imprinting and "mitotically inherited random monoallelic association" are explored experimentally.

7. lines 876 – 901. The addition of the Nanopore sequencing to corroborate data from Pac-Bio sequencing in the original submission. That said, the description of the statistical analyses in lines 876 – 901 is unclear and I would advise the authors to either change their statistical analysis or at least explain some of the things they are doing in a more convincing manner. It is not clear why multiplying the empirical p values is a good idea; such multiplying makes the ultimate p value be dependent on the number of things being multiplied. Using an Erlang distribution with the parameters they chose is not justified and is not a common way of analyzing sequencing data.

*Reviewer #2 (Recommendations for the authors):*

The manuscript presented by Hao, Liu, et al. details important findings surrounding a new type of noncoding RNA that regulated the expression of rRNA. The characterization of SNLU-1 through a variety of imaging and sequencing based approaches is compelling and provides a new example of noncoding RNAs that can control a major biological process in an allele-specific manner.

Summary of the work:

Ribosomal RNA regulation is of critical importance and therefore so too is defining the molecular mechanisms that control its expression. A long-standing question has been to understand which of the many nucleolar organizing regions (NORs) are active by what means the cell establishes this. Here the authors characterize a new, noncoding RNA based mechanism for regulation of rRNA expression. Leveraging confocal imaging and super resolution reconstructions, the authors characterize a new type of noncoding RNA called SNUL (focusing mostly on SNUL-1, but also presenting data on SNUL-2). These noncoding RNAs associate with and generate 'clouds' around chromosomes in an allele specific manner. Examining this across a number of cell lines at various cell cycle stages, the authors discovered that SNUL-1 is mitotically inherited in a random monoallelic association with chromosome 15. The SNUL RNAs are Pol I expressed and using long read sequencing on two different technology platforms provided some sequence context for the RNAs, with the caveat that they are difficult to map fully due to their repetitive nature. Finally, knockdown studies revealed an impact on rRNA processing through imaging and northern blotting assays, highlighting a functional consequence of SNUL transcripts.

Overall, the claims in the paper are well supported by the data presented. The revised manuscript also extensively investigates a previously reported observation of reported allelic-specific replication and transcription of rDNA loci in human cells. The new data generated here, with newer techniques appears to be internally consistent, while contradictory to the previous report. Beyond rRNA biology, the phenological observation of the cloud-like association of the SNUL's with various chromosomes is an exciting new example of a noncoding RNA establishing a physical territory around a specific chromosome to control the expression in situ.

There are some additional data presentation aspects and experiments that could add to the depth of understanding. Some of these would likely be the focus of future studies.

1. Visualizing the SNUL transcripts. A northern blot could have been performed to give readers a sense of the length and abundance of the various isoforms. If this information was also obtained from the Nanopore or PacBio data, the authors could have compiled it into a figure / plot to again provide this information to the readers.

2. There are some places where the authors note "data not shown". Generally, I would avoid this, if it is important enough to mention in the text, the data should be shown. If not, there is no need to note the data in the text.

3. It was not obvious how long the authors waited after KD in Figure 1-S2J. The figure shows loss of SNUL-1, but no data is shown about the effect of this KD on the pre-rRNA, even though the text says there was no effect on the pre-rRNA. Additional information here or data on the pre-rRNA would be helpful.s

Future experiments:

• In the future, an RNA-capture assay should be performed. Assaying both the associated RNA, DNA, and proteins using ChART, ChIRP, RAP, or OMAP technologies would likely provide mechanistic insight into how and importantly where, at a molecular level, the SNULs operate.

Other comments on the text and figures:

• Figure 1B; why do U2OS have such a speckly pattern of SNUL-1?

• Figure 1C; the labels / legends have squares overlapping the text.

• Figure 2I and related; the Pol1 data is convincing, but I am not sure that I see an obvious shift to the periphery of the nucleoli. Maybe illustrate where the nucleoli are in some of these images?

• Figure 5G; is there a way to quantify the difference here and apply a statistical test to demonstrate significance?

• All of the Red/Green combos are not ideal for data presentation in the case of color-blind readers. There are also higher contrast 2 and 3 color sets that can make the data easier to read for everyone.

*Reviewer #3 (Recommendations for the authors):*

This revision by Hao and Liu et al., adds an extensive confirmation study of Schlesinger et al. 2009 Genes and Dev., which reported asynchronous replication timing and mono-allelic expression of human nucleoli-organizing regions (NORs). However, neither observation is reproduced here, which is noteworthy given the original study has been extensively cited but to my knowledge not been reproduced. These contradictory results probably warrant publication on their own.

Overall, this revision is highly responsive to prior reviewers comments, and technically rigorous. Unfortunately, a single critical point remains unresolved: the location(s) of SNUL-1 and to which extent its effect on pre-rRNA processing can be distinguished from targeting the pre-rRNA itself:

The SNUL1 probe does not map to any NOR-carrying chr by blat. Because the SNUL candidate sequences (CS) or multiple-sequence alignment were not included in this revision, their alignment to the T2T genome (Nurk, et al. 2022) could not be assessed. At a minimum, the authors should include this information in a revision.

The SNUL2 probe does map to all NOR-chromosomes however, with multiple blat hits in the rDNA and one additional partial hit outside each rDNA cluster. On chr13 it is immediately adjacent to the core rDNA cluster (see blat to T2T genome on UCSC browser). Given the overlapping CT-dinucleotide repeat region common to SNUL-1/2, it is noteworthy that the only ASO capable of reducing SNUL-1 also reduced SNUL-2 by targeting the shared CT-dinucleotide repeat region (ASO sequence was not provided). In addition to the SNUL1-CS sequence alignments mentioned above, the authors should also provide FISH data from ASO experiments against the SNUL-1 CS sites (currently listed as "data not shown").

Without this and the ASO information, it remains unclear to me whether the observed pre-rRNA processing defect in ASO-targeted cells isn't a result of targeting the pre-rRNA itself, given its extensive sequence homology with SNUL-1 CSs (Figure 1). Of course, the pre-rRNA signal increases in these cells, while the SNUL-1/2 signal disappears, but the ASO may interfere with SNUL-1/2 probe hybridization, which relies on the CT-dinucleotide sequence as well (Figure 1-suppl2). In contrast, the pre-rRNA is detected with a different probe. At minimum, the authors would need to demonstrate SNUL-1 depletion with a probe that does not overlap the ASO-targeted sequence, to demonstrate that the effect of the ASO on SNUL-1 is depletion, while the pre-rRNA signal increases.

In summary, this study identifies a novel class of nucleoli-associated non-coding RNAs associating in cis with a specific NOR (SNUL-1 on chr15 and SNUL-2 on chr13). In addition, this study makes a strong case against monoallelic rRNA expression reported previously.

The technical challenge of pinpointing the origin of these SNULs even with the recently T2T-assembled human rDNA repeats is well appreciated, but the authors need to provide additional information to clarify the identity of SNUL-1 CSs. To link SNUL-1 function to rRNA synthesis, the authors would have to disambiguate the SNUL-1 target from co-targeting the pre-rRNA itself – for example by combining multiple SNUL-1 CS ASOs without sequence homology to the pre-rRNA, or at minimum, using SNUL-1 CS FISH probes that do not overlap with the ASO to confirm SNUL-1 depletion. If the only ASO effective at "depleting" SNUL-1 also matches the pre-rRNA itself, the authors should test ASOs that target only the pre-rRNA – if a processing defect is reproduced here, it is unlikely a result of targeting SNUL-1 or other SNULs.

[Editors’ note: further revisions were suggested prior to acceptance, as described below.]

Thank you for resubmitting your work entitled "Monoallelically-expressed Noncoding RNAs form nucleolar territories on NOR-containing chromosomes and regulate rRNA expression" for further consideration by *eLife*. Your revised article has been evaluated by Kevin Struhl (Senior Editor) and a Reviewing Editor.

The reviewers are now very positive about your paper and we are ready to move forward. However, we would like you to discuss the 2 remaining points mentioned by Reviewers 2 and 3. No new experiments are required. A short discussion will do. Once we receive these revisions, we will be ready to proceed to publication.

*Reviewer #2 (Recommendations for the authors):*

I think that the authors for their work revising the manuscript and adding new key data to support their claims. I have better confidence in the conclusions made by the authors now. I would like to request one last item before publication. When re-reading the manuscript, I am still left with confusion about where the SNULs are coming from. I understand they have imperfect homology with specific parts of the pre-rRNA, and thus perfect alignment is not possible.

However, as a reader, I feel somewhat confused about where these RNAs are being transcribed from.

I think a model of the rRNA, annotated with the expected regions or regions of homology, in the context of the other known domains of the rDNA would be extremely helpful. For example, there is an opportunity to provide a singular resource and reference for readers with the cartoon illustrated in "Figure 1—figure supplement 3A" (Schematic showing the positions of the rRNA and ITS1 probes.) Is it possible to layer onto this all of the probes, ASOs, etc used across this work as well as the expected region of SNUL transcription – with the fair caveat that the precise region of transcription is not yet defined?

*Reviewer #3 (Recommendations for the authors):*

I want to commend the authors on thoroughly addressing the reviewers concerns..

I just don't understand how the SNUL sequences don't map to even the T2T genome. I've blat-ed several of the candidate sequences and the 86% identity matches often hit a particular rDNA model (e.g. supplementary file 2, candidate 1 aligns to "rdna-model15a", which would be congruent with the SNUL1 cloud lighting up a rRNA-adjacent area). Yet, looking at the mismatches in the blat, they are primarily guanosines and adenosines. Aside from a more in-depth statistical analysis of the underlying sequence content of the 21S rRNA transcript, have the authors considered the possibility that all SNUL-1 clouds originate from heavily A-I or other RNA-edited 18S/21S rRNAs?

RNA-editing of 18S/21S rRNA could account for the significant dissimilarity to 21S while 21S is still the best genomic match for these Iso-Seq and Nanopore sequences. This interpretation would suggest that the SNULs are retained in the nucleus rather than exported, to perform regulatory functions or because editing impaired export. If the authors cannot rule out RNA-editing (e.g. would ADAR knockdown impair SNUL1 clouds?), could they discuss this possibility in the manuscript?

A cursory literature search revealed there's precedence for ADARs associating with nucleoli (Sansam, et al. 2003, PNAS – 223 citations).

https://www.ncbi.nlm.nih.gov/pmc/articles/PMC283538/

To be clear, I don't think additional experiments are necessary, but the possibility that SNUL clouds result from nucleolar retention of RNA-edited rRNA transcripts should be discussed. Lastly, I recommend checking the manuscript once more for typos (e.g. "Is-Seq") and sentence structure.

---

## [Author Response]

[Editors’ note: The authors appealed the original decision. What follows is the authors’ response to the first round of review.]

1. The reviewer(s) found the paper interesting, but the main concerns are that the study does not provide enough information about the RNA

We thank the editor for the comment. We have characterized the SNUL-1 ncRNA in the present study. To summarize, we have (1) identified the full-length sequence of SNUL1 by utilizing more than one long-read sequencing approach, (2) mapped the fine localization of SNUL-1 within the nucleolus by super-resolution imaging, (3) its cell line-specific maternal/paternal monoallelic association to Chromosome 15, and finally, (4) its potential involvement in regulating the expression and/or processing of rRNA genes. However, we were not able to map the genomic location of *SNUL1* genes on the repeat-rich acrocentric Chr.15 p-arm. This is a technical limitation, and we appreciate that the reviewer 1 also acknowledged this limitation, stating that “a clear determination of the genomic origin of these RNAs will be complicated by the repetitive sequences in the regions of the genome where they reside.”

2. and equally important, the manuscript did not discuss preceding work that impacts the interpretation and novelty of the findings. The observation of monoallelic expression from rRNA encoding loci is interesting, but it has been established in 2009 by the work, Allelic inactivation of rDNA loci. Genes Dev. 2009 Oct 15;23(20):2437-47. doi: 10.1101/gad.544509. You and your coauthors may not be aware of the paper, but there may be relevant data from this paper to help understand the NOR phenomenon better.

We thank the reviewing editor and the reviewer for pointing out the paper from the Cedar lab (Genes Dev. 2009 Oct 15;23(20):2437-47 ^1^). In this paper, the authors claimed allele-specific asynchronous replication of rDNA loci. The authors observed that half of the 10 rDNA loci in human cells replicated late during the S-phase, and the other half replicated early in an allelic-specific manner. It is generally presumed that late replicated DNA loci tend to be transcriptionally inactive. Due to the high sequence similarity between rRNAs transcribed from the rDNA repeats in all 10 NORs, the authors did not concisely demonstrate allele-specific transcription inactivation of rRNA genes. However, in support of their claim, they demonstrated that 5 of the 10 NOR containing acrocentric arms in the mitotic chromosomes stained weakly for active chromatin histone marks such as H3Ac, H4Ac, and H3K4me, implying that these chromosomes contain inactive rRNA repeats. Based on these results, the authors argued that rDNA foci display mono-allelic expression.

As per the editorial suggestion, we performed exactly the same experiments conducted by the Cedar lab in their 2009 Genes and Development paper^1^ to test the status of rDNA allele-specific replication and association of active histone marks on the mitotic rDNA cluster-containing p-arm of Chr-15 in three different cell lines (WI-38, HTD114 and RPE1) those show a mono-allelic association of SNUL1. Furthermore, we performed the following experiments to determine whether SNULs play any role in the allele-specific asynchronous replication and transcription repression of rRNA loci.

We performed the ReTiSH (Replication timing-specific hybridization) in control and SNUL-depleted cells (HTD114 and WI-38). Unlike what Cedar group reported earlier, we DID NOT observe allele-specific differential replication of rDNA clusters from chr.15 p-arm. Rather, we observed that both the alleles of Chr.15 p-arms replicate during the late S phase in both the cell lines (Figure 5 – Supplement3; Figure 5 – Supplement 4). Interestingly, upon SNUL depletion, the rDNA clusters from two Chr.15 alleles showed differential replication timing in HTD114 cells. The Maternal allele of Chr.15 p-arm replicates earlier (during mid S-phase) than the paternal allele (late S-phase) in SNUL-depleted cells (Figure 5 – Supplement3).To confirm the ReTiSH results, we performed EdU pulse-chase experiments in control and SNUL-depleted interphase cells. In these experiments, cells were pulse-labeled with EdU to mark replication sites, then co-stained with rDNA and Chr.15-specific centromeres to determine the replication timing of the Chr.15 p-arm containing rDNA cluster (Figure 5 – Supplement 5). Similar to our ReTiSH data, in control cells, rDNA cluster from both the chr-15 p-arms preferentially associated with EdU +ve foci in cells showing mid and late S-phase patterns. Upon SNUL depletion, one of the rDNA clusters from Chr. 15 showed enhanced association with EdU foci in middle S-phase cells (Figure 5 – Supplement 5). These results implicate that the replication timing of maternal allele shifts from late S-phase to middle S-phase upon SNUL depletion, confirming the ReTiSH results. The above results indicate that SNULs influence the replication timing of rDNA regions.

However, we could not reproduce the Cedar lab’s results showing allele-specific differential replication of rDNA loci in control cells.

Next, we tested the mono-allelic association of three of the active histone marks (H3Ac, H4Ac, H3K4me3) on the rDNA cluster-containing p-arms of Chr.15. We performed these experiments in control and SNUL-depleted HTD114 and RPE1 cells (Figure 5 – Supplement 6; Figure 5 – Supplement 7), again both these cells showed allele-specific expression of SNUL-1. In addition, SNULs influence the expression of rRNA genes in these cells. We observed active histone mark signals in all the chromosome arms, including the q-arms of chr.15. However, unlike what had been reported earlier by the Cedar lab in their Genes and Dev. paper, we could NOT find any active histone marks in both the alleles of rDNA bearing p-arms of chr.15. We measured the histone mark signal intensities between two chr15 p-arm alleles, and we could not find any significant difference between two alleles in control and SNUL-depleted cells (Figure 5 – Supplement 6; Figure 5 – Supplement 7).

These results further argue against the model proposed by Cedar lab supporting mono-allelic transcription of rDNA genes.

In support of their claim, Cedar lab study also showed allele-specific DNA CpG methylation (a mark associated with inactive transcription) on rDNA clusters. However, recent DNA methylation data derived from long-read sequencing data support the hypothesis that each rDNA array is a mosaic of active rDNA repeat and silent rDNA repeat clusters. These studies clearly demonstrated that all the NORs contain both methylated and unmethylated 47S rDNA clusters, and the methylated rDNA repeats within each NOR tend to cluster together to form heterochromatic regions ^2, 3^.Further, the UBF, an RNA polymerase I transcription factor, is associated with the transcriptionally active rDNA repeats ^2-4^. It is now established that in diploid human cells, all of the rDNA repeat containing NORs associate with UBF, indicating that all the 10 NOR alleles contain active rDNA clusters ^2-5^. These observations further contradict the Cedar lab study.

Based on our experiments performed in three different diploid or pseudo-diploid cell lines (WI-38, RPE1, and HTD114), we could not reproduce the data published by the Cedar lab. We did not see allele-specific differential replication of rDNA cluster located in Chr.15 p-arm. We also did not see any active histone marks on both the alleles of rDNA-cluster from the Chr.15 p-arm, implying the absence of allele-specific transcription of rDNA clusters. We also would like to point out that we have not seen any other study (including from Cedar lab) that followed (or reproduced) the original study from Cedar lab, demonstrating allele-specific transcription of rDNA genes. The consensus in the field is that all rDNA-positive NORs are a mosaic of both transcriptionally active (hypomethylated, UBF-associated) and inactive (hypermethylated, UBF-negative) rDNA repeats ^5^. The important question that remains to be addressed is how differential expression of rDNA repeats within each NOR cluster is achieved. SNULs could be one of those ncRNAs, regulating differential rDNA expression within each allele.

Based on all these observations, we therefore argue that our study, showing the mono-allelic association of SNUL-1 and -2 transcripts to Chr.15 and.13 respectively constitute the first study demonstrating the allele-specific association of ncRNAs on the rDNA-cluster containing NORs.

3. We do acknowledge that you have succeeded in identifying and characterizing a new family of ncRNAs which seem to play a role in the mono-allelic expression of r-RNA. These ncRNAs are expressed in a mono-allelic manner. This remains interesting but we feel that the study would have to go further since monoallelic expression was established more than 10 years ago.

We thank the reviewer for the encouragement. As indicated above (Comment 2), the Cedar lab paper did not demonstrate allele-specific transcription of rRNA transcripts due to technical limitations. They argued the mono-allelic inactivation of rDNA genes based on their asynchronous replication status and the low active chromatin marks on half of the rDNA loci in mitotic chromosomes. However, we could not reproduce their data in three different human cell lines. Also, recent studies have demonstrated data indicating that every rDNA repeat-containing NOR constitutes both active and inactive rDNA repeats. Such data contradicts the Cedar lab model, claiming that only half of the NOR-positive acrocentric p-arms contain active rDNA clusters.

Based on this, we argue that our observations, showing mono-allelic association of SNUL-1 and -2 ncRNAs to NOR-containing chromosomes is a novel finding. We also demonstrated that SNULs represent a novel class of ncRNAs, forming distinct nucleolar territories on NOR-containing chromosomes. Though SNUL-1 ncRNA displayed sequence similarity to 21S pre-rRNAs, it does not represent 21S pre-rRNA variants. Our recent unpublished data demonstrates that SNUL-1 ncRNAs are NOT processed into distinct 18S rRNA variants and do not associate with 40S ribosomes (data not shown).

4. Reviewer 1 also mentioned that the experiments would have to be better described for them to evaluate the findings.

We apologize for not explaining the experiments clearly. We have now modified the text in the revised version and have highlighted it in the revised manuscript.

5. We hope these comments will be helpful to you as you move forward. We hope that you will conduct further experiments that would address the above concerns.

We thank you for pointing out the issues. Our experiments clearly demonstrated SNUL family ncRNAs represent a novel family of ncRNAs that display monoallelic association to specific rDNA cluster-containing p-arms and regulate rRNA expression.

Recommendations for the authors:The manuscript "Monoallelically-expressed Noncoding RNAs form nucleolar territories on NOR-containing chromosomes and regulate rRNA expression" reports the discovery of a family of ncRNAs they call SNULs for Single NUcleolus Localized RNA and examine their localization with respect to nucleoli and reports that the RNAs they are examining are monoallelically expressed in a mitotically stable manner similar to what happens in X inactivation.These RNAs come from a screen which is not well described and the descriptions of the sequence analyses are unclear, so it is difficult to know exactly what they are analyzing in the manuscript.

We apologize for not including the required details in the manuscript.

The cell cycle lncRNA screen where we identified the initial SNUL-1 probe was published in an earlier paper ^6^. By performing RNA-seq in cell cycle synchronized samples, we identified several hundreds of lncRNAs that differentially expressed in a particular stage of the cell cycle. We performed a large-scale RNA-FISH-based screen to characterize the localization of these cell cycle-regulated lncRNAs. One of the probes in this screen hybridized to SNUL-1 RNA in the nucleolus. The original double-stranded DNA probe that detected the SNUL-1 RNA cloud(s) was mapped to hg38-Chr17: 39549507-39550130 genomic region, encoding a lncRNA. However, other unique non-overlapping probes generated from the Chr17-encoded lncRNA failed to detect the SNUL-1 RNA cloud. Furthermore, BLAST-based analyses failed to align the SNUL-1 hybridized sequence to any other genomic loci. Since a large proportion of the p-arms of nucleolus-associated NOR-containing acrocentric chromosomes is not yet annotated, we speculated that SNUL-1 could be transcribed from an unannotated genomic region from the acrocentric p-arms.

We have now provided the information in the revised manuscript. Specifically, we have provided the details of the PacBio iso-seq, nanopore seq analyses as well as the bioinformatic approaches that were conducted to determine the identity of the full-length SNUL-1 ncRNA.

If these are RNAs with reasonable abundance, then they should be findable without the extensive PCR amplification they appear to have done for the PacBio sequencing (the methods section is not clear on exactly how many rounds of PCR were performed).

We apologize for not providing the essential details. In the PacBio-iso-seq analyses, we utilized the standard protocol (recommended by the scientists from PacBio, who are authors in the manuscript), which included 13 PCR cycles. However, as described in the manuscript, in parallel to PacBio-seq, we also performed nano-pore sequencing of the nucleolus-enriched RNA without any amplification. The SNUL-1 full-length candidate sequence (CS) that we described in the manuscript is the ncRNA that showed 100% sequence similarity in both independent PacBio Iso-seq as well as nanopore seq analyses. We argue that if the SNUL-1 candidate transcripts would have been an artifact of PCR amplification in PacBio-seq, then we would not have obtained the full-length sequence with 100% match in the nanopore-seq reads. We have now included the detailed bioinformatic analyses in the methods section of the manuscript.

Moreover, given the acknowledged sequence similarities of the SNULs with other RNAs, the possibility of chimaera formation during PCR amplification is high. They are clearly detecting RNAs associated with nucleoli but exactly what they are examining is unclear.

Please see our response above. In addition, we performed detailed bioinformatic analyses to test whether the SNUL-1 full-length sequence obtained in the PacBio-seq is not an artifact of PCR amplification. This analysis is described in detail in the methods section under the sub-title “sequencing analyses”.

It is possible that a clear determination of the genomic origin of these RNAs will be complicated by the repetitive sequences in the regions of the genome where they reside.

We thank this reviewer for acknowledging the technical limitation of mapping the genomic locus of SNUL1 genes. We have pointed out this as the limitation of the present manuscript. Mapping the *SNUL-1* genomic locus and characterizing the regulatory sequence elements and factors that control the monoallelic expression of SNULs will be part of future research plans.

Note also that the idea of monoallelic expression from rRNA encoding loci is interesting, but has been established in 2009. Title: Allelic inactivation of rDNA loci. Genes Dev. 2009 Oct 15;23(20):2437-47. doi: 10.1101/gad.544509.

We thank the reviewer for pointing out the study from Cedar lab published in 2009. To test the idea that SNULs contribute to allele specific expression of rRNA, which was previously reported by Cedar lab in their 2009 GandD paper, we performed the same set of experiments described in their paper in three different cell lines in the presence or absence of SNULs (please see the response to Comment 2). However, we could not reproduce any of the data presented in the GandD manuscript. Also, we have not seen any other follow up study, where mono-allelic expression of rDNA genes was observed. Currently, no concrete data supports monoallelic expression of rRNA ^5^. We, therefore, argue that our current study is the first one, demonstrating the mono-allelic association of a ncRNA from the p-arm containing rDNA cluster.

If there are analyses which could be added to further clarify how the different types of sequencing complemented and/or reinforced one another, that might make the characterization of the RNAs being studied here more convincing.

We have performed extensive analyses to identify the full-length SNUL-1 sequence. We have included the details in the results and methods section with better clarity.

Further data added in the revised manuscript supporting the role of SNULs in the nucleolar organization and rRNA expression.

By performing single-molecule RNA FISH, we demonstrated that the depletion of SNULs enhanced the levels of pre-rRNAs (Figure 5 – Supplement 1C-F). We further validated the RNA-FISH data on individual cells by using a modified flow cytometry-based assay by which we showed increased pre-rRNA levels in a significant population of SNUL-depleted cells (Figure 5 – Supplement 1G).We observed that SNULs influenced the organization of sub-nucleolar compartments. SNUL-depleted cells showed increased number of FC/DFC compartments/nucleolus (Figure 5D and F). In addition, upon SNUL depletion, the DFC unit becomes smaller in size, while FC shows no significant difference between control and SNUL-depleted cells (Figure 5G).

We hope the reviewers would find our manuscript suitable for publication in *eLife*.

Reference

1. Schlesinger, S., Selig, S., Bergman, Y. and Cedar, H. Allelic inactivation of rDNA loci. *Genes and development* 23, 2437-2447 (2009).

2. Hori, Y., Shimamoto, A. and Kobayashi, T. The human ribosomal DNA array is composed of highly homogenized tandem clusters. *Genome research* 31, 1971-1982 (2021).

3. van Sluis, M., van Vuuren, C., Mangan, H. and McStay, B. NORs on human acrocentric chromosome p-arms are active by default and can associate with nucleoli independently of rDNA. *Proceedings of the National Academy of Sciences of the United States of America* 117, 10368-10377 (2020).

4. van Sluis, M., van Vuuren, C. and McStay, B. The Relationship Between Human Nucleolar Organizer Regions and Nucleoli, Probed by 3D-ImmunoFISH. *Methods in molecular biology (Clifton, N.J.)* 1455, 3-14 (2016).

5. McStay, B. The p-Arms of Human Acrocentric Chromosomes Play by a Different Set of Rules. *Annual review of genomics and human genetics* (2023).

6. Hao, Q. *et al.* The S-phase-induced lncRNA SUNO1 promotes cell proliferation by controlling YAP1/Hippo signaling pathway. *eLife* 9 (2020).

[Editors’ note: what follows is the authors’ response to the second round of review.]

Specifically, we hope you can address these comments in a revised paper.1. Reviewer 2 suggested visualizing the SNUL transcripts using a northern blot to give readers a sense of the length and abundance of the various isoforms. If you can provide this information from the Nanopore or PacBio data, please do so in lieu of the northern. (also see note from Reviewer 1 about presentation of the Nanopore sequencing data)

The SNUL-1 candidate transcripts show ~84-90% overall sequence similarity with the 21S pre-rRNAs. Because of that, the probes in Northern blot using SNUL probes primarily detected highly abundant pre-rRNAs. In the revised manuscript, as per the reviewer’s suggestion, we have now provided the pac-bio-seq/nanopore seq data showing the complete alignment of Nanopore reads on the full-length SNUL-1 candidates identified by targeted iso-sequencing (Figure 1 – Supplement 2C). Please also see Supplementary file 2 for the full-length SNUL-1 CS sequences, identified by the targeted Pac-Bio seq.

2. Reviewer 3 wished to see the location(s) of SNUL-1 and to which extent its effect on pre-rRNA processing can be distinguished from targeting the pre-rRNA itself. Please see what info you could provide in a revision.3. Disambiguation of the SNUL-1 target from co-targeting the pre-rRNA itself.

We thank the reviewer for his/her/their suggestion. As suggested by this reviewer, we have now performed Northern blots to detect the changes in pre-rRNA processing after SNUL or pre-rRNA depletion. To deplete pre-rRNA, we used Antisense oligonucleotides targeting 18S rRNA as well as the ITS1 within the 21S pre-rRNA. Our results clearly demonstrate that only SNUL, and not pre-rRNA depletion affected the levels of 30S pre-rRNA (Figure 5-Supplement 2C). These results imply that changes in 30S pre-rRNA levels observed in SNUL-ASO treated cells is specifically due to SNUL depletion.

To test the reviewer’s concern that absence of SNUL-1 signal in the SNUL-ASO-treated cells because of potential competition between SNUL-ASO and the probe, (both ASO and SNUL-1 probe competing for the SNUL-1 RNA), we performed the following experiment. We incubated cells with con (scr) and SNUL-1 ASO for 72 hrs. After that, we fixed cells, denatured the cells and performed high stringent washes (similar to striping the probes in MER-FISH protocol) for striping the ASOs from the target RNA. After that, we performed RNA-FISH to detect SNUL-1. Results revealed loss of SNUL-1 signal in the cells treated with SNUL-1 ASO (Figure 5-figure supplement 2I). If the idea that the lack of SNUL-1 signals in SNUL ASO treated cells is because SNUL ASO competes with the probe for SNUL-1 RNA is true, then upon denaturation of cells treated with SNUL ASOs, would have shown positive SNUL-1 RNA signal. Based on that, we suggest that SNUL-ASO indeed degrades SNUL RNAs.

4. Please also address the more specific and minor comments of the reviewers.

Please see below the response.

Reviewer #1 (Recommendations for the authors):The manuscript by Hao et al., "Monoallelically-expressed Noncoding RNAs form nucleolar territories on NOR-containing chromosomes and regulate rRNA expression" characterizes novel non-coding RNA emanating from the rDNA regions of the human genome. The revised manuscript has thoughtfully considered the concerns raised in the initial review.

We thank this reviewer for his/her/their comments.

Comments / points to be addressed:1. line 66/67, instead of "and were transcriptionally inactive" perhaps say something along the lines of "and had chromatin marks consistent with being transcriptionally inactive".

We have now modified the text as per the suggestion.

2. line 178/179. Please give a little more detail as to what you mean by "partially align to 18S" and "align to ITS1". What % alignment over what distance, or whatever makes sense given the nature of the alignments being mentioned?

We apologize for not explaining this clearly. We have now modified the text in the revised manuscript as follows:

“In general, the individual SNUL-1 CSs showed 84-86% alignment to the 5’ end of 21S and 81-90% alignment to the 3’ end of 21S (Figure 1F). For example, SNUL-1 CS2305 showed 86% identity to the first 1.3 kb of 21S pre-rRNA, which corresponds to a significant portion of 18S rRNA, followed by a large gap and segments of 86-91% identity corresponding to the 3’ end of 18S and the 3’ end of ITS1 (Figure 1F). On the other hand, CS1269 had 91% identity to the 18S rRNA region but had a large gap in alignment in the first 1 kb, as well as segments of 84-90% identity corresponding to the 5’ and 3’ ends of ITS1 (Figure 1F)”.

3. line 242. Use of the word "biallelically" is confusing in this context because it seems that the text is referring to a second location which could be one of the two alleles of several rDNA loci.

We agree with the reviewer’s comment. We do not have any direct evidence of biallelic expression of SNUL1 in telophase/early G1 nuclei. We therefore have toned down that sentence and described that “during telophase/early G1 nuclei, SNUL1 is expressed from two independent loci”.

4. lines 251 -258. The text is talking about the SNUL-1 cloud association with one allele of Chr. 15 NOR and then mentions the association rate with Chr. 13 and 22, but not the other allele of Chr. 15.

Co-RNA-DNA FISH to detect SNUL-1 RNA and Chr.15 in cells revealed that in 100% of the interphase nuclei, the single SNUL-1 cloud associated only with one of the two chr 15 alleles (please see figures 3B, C). This is also evident from the data presented in Figures 4A and C, where in all the cells, SNUL-1 cloud only associated with one of the two chr 15 alleles, marked by chr. 15 specific centromeres or peri-centromeres. These experiments were done in diploid cells, which normally contain 1-2 nucleoli. The association of SNUL-1 to a small percentage of other acrocentric chromosomes (besides Chr. 15) could be because these chromosomes are localized next to the SNUL-1-associated Chr. 15 allele in the same nucleolus, and not because SNUL-1 cloud is coating these chromosomes. We have discussed this point in the manuscript.

5. lines 285 – 287. What about mentioning if it could be observed whether the other Chr. 15 had transcriptionally active rDNA clusters?

We thank the reviewer for this suggestion. SNUL-1- decorated FC/DFC region contained pre-rRNA (Figure 3E). Also, SNUL-1-decorated Chr.15 were transcriptionally active as observed by positive 5-FU incorporation (Figure 3-supp.1J). Both these data support the models that SNUL-1-associated Chr. 15 allele contains active rDNA clusters. At present, we do not have any direct evidence to show whether the rDNA clusters within the SNUL-1 -ve Chr-15 allele remain active or inactive. This is a technical problem. We were not successful in performing super-resolution imaging of RNA-FISH (to detect pre-rRNA) and DNA paint (to detect chr. 15 alleles). However, indirect evidence supports the model that both alleles of Chr. 15 contain active rDNA genes. The presence of UBF, a protein associated with active rDNA clusters, in both the Chr.15 alleles (Figure 3-supp.1I) support the model that both the chr. 15 alleles contain active rDNA genes.

6. lines 300 – 303. I suggest not using "non-randomly associates" as this is vague, and instead use terminology similar to what is used in the next paragraph where the ideas of imprinting and "mitotically inherited random monoallelic association" are explored experimentally.

Thanks for the suggestion. We have modified this sentence accordingly.

7. lines 876 – 901. The addition of the Nanopore sequencing to corroborate data from Pac-Bio sequencing in the original submission. That said, the description of the statistical analyses in lines 876 – 901 is unclear and I would advise the authors to either change their statistical analysis or at least explain some of the things they are doing in a more convincing manner. It is not clear why multiplying the empirical p values is a good idea; such multiplying makes the ultimate p value be dependent on the number of things being multiplied. Using an Erlang distribution with the parameters they chose is not justified and is not a common way of analyzing sequencing data.

We have added a more detailed description of the statistical analysis performed and is included in the revised manuscript.

The reviewer is correct about the product of (empirical) p-values not being a well-calibrated indicator of significance. We have now performed the Fisher combined probability test (also called the Fisher’s method for aggregating p-values) to perform a rigorous statistical test on the product of p-values, obtaining a p-value of < 0.000002 from this test. Briefly, Fisher’s method combines a set of k p-values into the statistic χ2=−2∑iln(pi), which has a χ2 distribution with 2k degrees of freedom (for us, k=5 since five p-values are combined in this way). Note that this statistic is equivalent to multiplying the p-values, but assesses the significance of that product rigorously, ultimately providing us a p-value.

We have also added text explaining the use of the Erlang distribution and corrected a mistake in the previous text, where we had mistakenly quoted the value of the rate parameter rather than the scale parameter, which is inverse of the former. We also modified the analysis to use the Fisher’s method (combined probability test) and thus obtain a rigorous p-value used to reject the hypothesis that is called “H1” in the paper. The new text is reproduced below for the reviewer’s convenience.

“We also provide a rigorous statistical analysis to reject H1. First, we approximate the empirical distribution of the dissimilarity scores with an exponential probability density with mean 0.5%, i.e., 0.005. Second, we assume that the dissimilarity scores of the isoforms are independent and identically distributed (i.i.d.) random variables i.e., ∀i∈{1, …, 5}, DSi∼exp⁡(λ), λ=200 (since mean of exponential distribution is 1/λ). Given the i.i.d. assumption, the maximum pairwise dissimilarity for each pair of isoforms Ii and Ij, DSijmax=DSi+DSj, has Erlang distribution with shape parameter 2 and rate parameter 200, as the sum of two independent exponential random variables with the same rate parameter has an Erlang distribution with scale parameter equal to that common rate. This allows us to estimate a p-value for each of the (52)=10 observed pairwise dissimilarity scores DSij (Supplementary file 3). Subjecting the 10 resulting p-values to a Fisher’s combined probability test, we obtained a p-value < 1E-21 (χ2=154 df =20). This allowed us to reject the *H_1_* hypothesis, leading to accepting the *H_2_* hypothesis.”

Reviewer #2 (Recommendations for the authors):The manuscript presented by Hao, Liu, et al. details important findings surrounding a new type of noncoding RNA that regulated the expression of rRNA. The characterization of SNLU-1 through a variety of imaging and sequencing based approaches is compelling and provides a new example of noncoding RNAs that can control a major biological process in an allele-specific manner.Summary of the work:Ribosomal RNA regulation is of critical importance and therefore so too is defining the molecular mechanisms that control its expression. A long-standing question has been to understand which of the many nucleolar organizing regions (NORs) are active by what means the cell establishes this. Here the authors characterize a new, noncoding RNA based mechanism for regulation of rRNA expression. Leveraging confocal imaging and super resolution reconstructions, the authors characterize a new type of noncoding RNA called SNUL (focusing mostly on SNUL-1, but also presenting data on SNUL-2). These noncoding RNAs associate with and generate 'clouds' around chromosomes in an allele specific manner. Examining this across a number of cell lines at various cell cycle stages, the authors discovered that SNUL-1 is mitotically inherited in a random monoallelic association with chromosome 15. The SNUL RNAs are Pol I expressed and using long read sequencing on two different technology platforms provided some sequence context for the RNAs, with the caveat that they are difficult to map fully due to their repetitive nature. Finally, knockdown studies revealed an impact on rRNA processing through imaging and northern blotting assays, highlighting a functional consequence of SNUL transcripts.Overall, the claims in the paper are well supported by the data presented. The revised manuscript also extensively investigates a previously reported observation of reported allelic-specific replication and transcription of rDNA loci in human cells. The new data generated here, with newer techniques appears to be internally consistent, while contradictory to the previous report. Beyond rRNA biology, the phenological observation of the cloud-like association of the SNUL's with various chromosomes is an exciting new example of a noncoding RNA establishing a physical territory around a specific chromosome to control the expression in situ.

We thank the reviewer for his/her/their positive remarks.

There are some additional data presentation aspects and experiments that could add to the depth of understanding. Some of these would likely be the focus of future studies.1. Visualizing the SNUL transcripts. A northern blot could have been performed to give readers a sense of the length and abundance of the various isoforms. If this information was also obtained from the Nanopore or PacBio data, the authors could have compiled it into a figure / plot to again provide this information to the readers.

We thank the reviewer for his/her/their suggestion. We attempted to perform Northern blot (NB) using SNUL-1 probe. However, in NB, the probe cross hybridized to pre-rRNAs and therefore could not get a clear data. The SNUL1 transcripts show >83-90% similarity (all through the entire length of the transcript) with the 21S pre-rRNAs. We therefore could not generate a NB probe that specifically detects only SNULs.

As per the suggestion, we have included the SNUL-1 full-length sequence (supplementary file 2). As discussed in the methods, in the pac-bio seq, we enriched the transcripts complementary to the SNUL-1 probe by RNA pull down and long-read sequencing. Further bioinformatic analyses identified Top-ranked isoforms with high binding affinity with SNUL-1 Probe 4 as SNUL-1 candidate sequences (CSs) (Figure 1-supplement 2B). RNA-FISH using probes targeting each of these CS showed similar SNUL-1 could (Figure 1-supplement D-F), implying that the full-length transcripts that we identified by iso-seq represent SNUL-1.

When nanopore sequencing based reads were aligned to the full-length CSs generated by pac-bio, we observed the reads aligning throughout the transcript (Figure 1-supplement 2C).

2. There are some places where the authors note "data not shown". Generally, I would avoid this, if it is important enough to mention in the text, the data should be shown. If not, there is no need to note the data in the text.

We thank the reviewer for this suggestion. Please see the data below. We did not show these data in the manuscript because both the data sets gave negative results (Author response image 1 and Figure 5—figure supplement 1A). However, we will be happy to add the data as supplementary figures if the reviewer feels that it is important to include the data.

**Author response image 1. sa2fig1:** RNA-FISH (green) using unique probes designed from the chr. 17 RNA (without including the repeat) in WI-38 cells showed the absence of SNUL-1 cloud in the nucleolus.

3. It was not obvious how long the authors waited after KD in Figure 1-S2J. The figure shows loss of SNUL-1, but no data is shown about the effect of this KD on the pre-rRNA, even though the text says there was no effect on the pre-rRNA. Additional information here or data on the pre-rRNA would be helpful.s

We apologize the reviewer for this confusion. SNUL-1 KD (Figure 1-S2J) was performed for 48 hrs. RNA-FISH using ITS1 probe demonstrated a marginal increase in the levels of pre-rRNA upon SNUL depletion as observed in this figure. ITS1 probe will detect nascent as well as partially processed pre-rRNAs. Our later experiments (shown in Figure 5), using probes targeting 5’ETS region that only detects nascent pre-rRNAs clearly demonstrated upregulation of nascent pre-rRNA upon SNUL depletion.

The scope of Figure 1-S2J was to demonstrate that SNUL ASO specifically reduced only SNUL,.

We have now modified the text accordingly.

Future experiments:• In the future, an RNA-capture assay should be performed. Assaying both the associated RNA, DNA, and proteins using ChART, ChIRP, RAP, or OMAP technologies would likely provide mechanistic insight into how and importantly where, at a molecular level, the SNULs operate.

We thank the reviewer for this suggestion. Future experiments will be targeted to identify the SNUL-1-interacting chromatin and proteome to gain insights into the mode of action of SNULs.

Other comments on the text and figures:• Figure 1B; why do U2OS have such a speckly pattern of SNUL-1?

We apologize for the quality of the figure. We have now replaced the figure with a better quality one (Figure 1B). U2OS cells in general, contain a greater number of nucleolui (average number is 6, please see Figure 1C). SNUL-1 probe give more nuclear dots in U2OS nuclei compared to all other cells tested.

• Figure 1C; the labels / legends have squares overlapping the text.

Thanks for pointing out the error. We have now fixed this.

• Figure 2I and related; the Pol1 data is convincing, but I am not sure that I see an obvious shift to the periphery of the nucleoli. Maybe illustrate where the nucleoli are in some of these images?

Thanks for the suggestion. We have now made inset of a single nucleolus in Figure 2I and Figure 2-supplement 1F. It is known that upon RNA pol II inhibition, rRNAs redistribute around the nucleoli and form necklace-like structures (PMIDs: 20965417; 8612682)**.** We observed that both SNULs and rRNA show similar behaviors upon RNA pol II inhibition.

• Figure 5G; is there a way to quantify the difference here and apply a statistical test to demonstrate significance?

We have now added quantification to this data set (Figure 5H).

• All of the Red/Green combos are not ideal for data presentation in the case of color-blind readers. There are also higher contrast 2 and 3 color sets that can make the data easier to read for everyone.

We sincerely apologize for using red/green color combination. All the image acquisition and analyses were done by the previous graduate student, who graduated a year back. These images (Super-resolution) are processed in the University microscopy facility. There is a long que for to access those workstations to reprocess the data sets. The revision of the manuscript was done with the partial help of an undergraduate student and another graduate student. During the revision, we prioritize the efforts to complete the wet-lab and bioinformatic experiments that were requested by the reviewers. Also, several of the figures contains three colors.

In the future, we will generate the figures as per the reviewer’s suggestion, and we thank this reviewer for the suggestion.

Reviewer #3 (Recommendations for the authors):This revision by Hao and Liu et al., adds an extensive confirmation study of Schlesinger et al. 2009 Genes and Dev., which reported asynchronous replication timing and mono-allelic expression of human nucleoli-organizing regions (NORs). However, neither observation is reproduced here, which is noteworthy given the original study has been extensively cited but to my knowledge not been reproduced. These contradictory results probably warrant publication on their own.

We sincerely thank this reviewer for his/her/their encouraging comments. It took several months for us to repeat these experiments in multiple cell lines. At the end of the day, it was frustrating to know that we could not reproduce any of the data published in the previous study.

Overall, this revision is highly responsive to prior reviewers comments, and technically rigorous. Unfortunately, a single critical point remains unresolved: the location(s) of SNUL-1 and to which extent its effect on pre-rRNA processing can be distinguished from targeting the pre-rRNA itself:The SNUL1 probe does not map to any NOR-carrying chr by blat. Because the SNUL candidate sequences (CS) or multiple-sequence alignment were not included in this revision, their alignment to the T2T genome (Nurk, et al. 2022) could not be assessed. At a minimum, the authors should include this information in a revision.

We have included the SNUL-1 candidate sequences identified independently by iso-seq and Nanopore sequencing (Supplementary file 2). We have also included the Nanopore seq. alignment data of several of the SNUL-1 candidate sequences (Figure 1-supplement 2C).

T SNUL-1 candidate sequences did not map to T2T genome, The closest BLAT hit obtained from T2T genome was as follows:

**Author response table 1. sa2table1:** 

QUERY	nucDNA1_PK_combo__HQ_transcript/2305
SCORE	1743
START	2
END	2523
QSIZE	3003
IDENTITY	88.4%
CHROM	chr13
STRAND	+
Ref_START	9250197
Ref_END	9252367
SPAN	2171

We will be happy to include this information in the manuscript, if the reviewer suggests.

The SNUL2 probe does map to all NOR-chromosomes however, with multiple blat hits in the rDNA and one additional partial hit outside each rDNA cluster. On chr13 it is immediately adjacent to the core rDNA cluster (see blat to T2T genome on UCSC browser). Given the overlapping CT-dinucleotide repeat region common to SNUL-1/2, it is noteworthy that the only ASO capable of reducing SNUL-1 also reduced SNUL-2 by targeting the shared CT-dinucleotide repeat region (ASO sequence was not provided). In addition to the SNUL1-CS sequence alignments mentioned above, the authors should also provide FISH data from ASO experiments against the SNUL-1 CS sites (currently listed as "data not shown").

We developed several ASOs targeting individual SNUL-1 candidates. Unfortunately, none of the ASOs gave significant knock down. (Please see Figure 5-figure supplement 2A). We believe that SNUL-1 consists of a group of very similar, but not identical transcripts, originating from a Chr. 15 p-arm cluster. Iso-seq and Nanopore seq. data supported this claim. Knocking down one or few of the candidates may not make a significant impact.

Without this and the ASO information, it remains unclear to me whether the observed pre-rRNA processing defect in ASO-targeted cells isn't a result of targeting the pre-rRNA itself, given its extensive sequence homology with SNUL-1 CSs (Figure 1). Of course, the pre-rRNA signal increases in these cells, while the SNUL-1/2 signal disappears, but the ASO may interfere with SNUL-1/2 probe hybridization, which relies on the CT-dinucleotide sequence as well (Figure 1-suppl2). In contrast, the pre-rRNA is detected with a different probe. At minimum, the authors would need to demonstrate SNUL-1 depletion with a probe that does not overlap the ASO-targeted sequence, to demonstrate that the effect of the ASO on SNUL-1 is depletion, while the pre-rRNA signal increases.

We completely agree with the reviewer’s interpretation. Unfortunately, we do not have any probe that successfully hybridizes to SNUL-1 without the repeats. Even the probes that hybridizes to SNUL-1 CS shown in Figure 1-Figure supp. 2C-D also contain CT repeats.

To test the competition model (both ASO and SNUL-1 probe competing for the SNUL-1 RNA), we performed the following experiment. We incubated cells with con (scr) and SNUL-1 ASO for 72 hrs. After that, we fixed cells, denatured the cells and performed high stringent washes (similar to striping the probes in MER-FISH protocol) for striping the ASOs from the target RNA. After that we performed RNA-FISH to detect SNUL-1. Results revealed loss of SNUL-1 signal in the cells treated with SNUL-1 ASO (Figure 5—figure supplement 2I). If the idea that the lack of SNUL-1 signals in SNUL ASO treated cells is because SNUL ASO competes with the probe for SNUL-1 RNA is true, then upon denaturation SNUL ASO from the target RNA would have shown positive SNUL-1 RNA signal. Based on that, we suggest that even though the competition could still be a possibility, it cannot explain the complete loss of SNUL-1 RNA signal in the ASO-treated samples.

In summary, this study identifies a novel class of nucleoli-associated non-coding RNAs associating in cis with a specific NOR (SNUL-1 on chr15 and SNUL-2 on chr13). In addition, this study makes a strong case against monoallelic rRNA expression reported previously.

We thank for this reviewer for recognizing the strength and the novelty of the study

The technical challenge of pinpointing the origin of these SNULs even with the recently T2T-assembled human rDNA repeats is well appreciated, but the authors need to provide additional information to clarify the identity of SNUL-1 CSs. To link SNUL-1 function to rRNA synthesis, the authors would have to disambiguate the SNUL-1 target from co-targeting the pre-rRNA itself – for example by combining multiple SNUL-1 CS ASOs without sequence homology to the pre-rRNA, or at minimum, using SNUL-1 CS FISH probes that do not overlap with the ASO to confirm SNUL-1 depletion. If the only ASO effective at "depleting" SNUL-1 also matches the pre-rRNA itself, the authors should test ASOs that target only the pre-rRNA – if a processing defect is reproduced here, it is unlikely a result of targeting SNUL-1 or other SNULs.

As per the suggestion, in order to test whether the changes in RNA processing observed in SNUL ASO-treated cells is not due to the fact that this ASO is also targeting pre-rRNA, we treated cells with specific ASOs targeting the pre-rRNAs (ASOs targeting 18S rRNA and the ITS1 of pre-rRNA). Northern blot using the 5’ETS probe revealed that only SNUL, and not pre-rRNA depletion resulted in significant alteration in the 30S pre-rRNA (Figure 5-supplement 2C). These results imply that the changes in pre-rRNA processing observed in SNUL ASO-treated cells is not due to the reason that ASO is also co-targeting pre-rRNA.

[Editors’ note: what follows is the authors’ response to the third round of review.]

The reviewers are now very positive about your paper and we are ready to move forward. However, we would like you to discuss the 2 remaining points mentioned by Reviewers 2 and 3. No new experiments are required. A short discussion will do. Once we receive these revisions, we will be ready to proceed to publication.

We thank the editor for his/her/their positive comment. We have now modified the manuscript as per reviewers’ suggestions.

Reviewer #2 (Recommendations for the authors):I think that the authors for their work revising the manuscript and adding new key data to support their claims. I have better confidence in the conclusions made by the authors now.

We thank the reviewer for the constructive comments.

I would like to request one last item before publication. When re-reading the manuscript, I am still left with confusion about where the SNULs are coming from. I understand they have imperfect homology with specific parts of the pre-rRNA, and thus perfect alignment is not possible.However, as a reader, I feel somewhat confused about where these RNAs are being transcribed from.I think a model of the rRNA, annotated with the expected regions or regions of homology, in the context of the other known domains of the rDNA would be extremely helpful. For example, there is an opportunity to provide a singular resource and reference for readers with the cartoon illustrated in "Figure 1—figure supplement 3A" (Schematic showing the positions of the rRNA and ITS1 probes.) Is it possible to layer onto this all of the probes, ASOs, etc used across this work as well as the expected region of SNUL transcription – with the fair caveat that the precise region of transcription is not yet defined?

We thank this reviewer for the suggestion. We have now modified Figure 1—figure supplement 3A and have incorporated the points suggested by the reviewer. We have also discussed these points in the manuscript.

Reviewer #3 (Recommendations for the authors):I want to commend the authors on thoroughly addressing the reviewers concerns.

We thank the reviewer for the constructive comments.

I just don't understand how the SNUL sequences don't map to even the T2T genome. I've blat-ed several of the candidate sequences and the 86% identity matches often hit a particular rDNA model (e.g. supplementary file 2, candidate 1 aligns to "rdna-model15a", which would be congruent with the SNUL1 cloud lighting up a rRNA-adjacent area). Yet, looking at the mismatches in the blat, they are primarily guanosines and adenosines. Aside from a more in-depth statistical analysis of the underlying sequence content of the 21S rRNA transcript, have the authors considered the possibility that all SNUL-1 clouds originate from heavily A-I or other RNA-edited 18S/21S rRNAs?RNA-editing of 18S/21S rRNA could account for the significant dissimilarity to 21S while 21S is still the best genomic match for these Iso-Seq and Nanopore sequences. This interpretation would suggest that the SNULs are retained in the nucleus rather than exported, to perform regulatory functions or because editing impaired export. If the authors cannot rule out RNA-editing (e.g. would ADAR knockdown impair SNUL1 clouds?), could they discuss this possibility in the manuscript?A cursory literature search revealed there's precedence for ADARs associating with nucleoli (Sansam, et al. 2003, PNAS – 223 citations).https://www.ncbi.nlm.nih.gov/pmc/articles/PMC283538/To be clear, I don't think additional experiments are necessary, but the possibility that SNUL clouds result from nucleolar retention of RNA-edited rRNA transcripts should be discussed.

We thank this reviewer for this outstanding comment. We completely agreed with the reviewer that SNUL candidates still show the highest similarity to 21S rRNAs. As rightly suggested by the reviewer, SNULs could represent rRNA variants due to extensive post-transcriptional modifications, such as A-to-I editing. We have now discussed this aspect in the revised manuscript.

In the future, we will analyze the RNA-seq data and will perform other experiments to test the idea whether SNULs undergo A-to-I editing. Also, we will test the role of ADARs in the nuclear-retention and nuclear-localization of SNULs.

Lastly, I recommend checking the manuscript once more for typos (e.g. "Is-Seq") and sentence structure.

Thank you for your suggestion. We have corrected the typos to the best of our ability.